

# Twisted and untwisted negativity spectrum of free fermions

Hassan Shapourian[1], Paola Ruggiero[2], Shinsei Ryu[1] and Pasquale Calabrese[2,3]

**1** James Franck Institute and Kadanoff Center for Theoretical Physics,
University of Chicago, IL 60637
**2** SISSA and INFN, via Bonomea 265, 34136 Trieste, Italy
**3** International Centre for Theoretical Physics (ICTP), I-34151, Trieste, Italy

## Abstract

A basic diagnostic of entanglement in mixed quantum states is known as the positive partial transpose (PT) criterion. Such criterion is based on the observation that the spectrum of the partially transposed density matrix of an entangled state contains negative eigenvalues, in turn, used to define an entanglement measure called the logarithmic negativity. Despite the great success of logarithmic negativity in characterizing bosonic many-body systems, generalizing the operation of PT to fermionic systems remained a technical challenge until recently when a more natural definition of PT for fermions that accounts for the Fermi statistics has been put forward. In this paper, we study the many-body spectrum of the reduced density matrix of two adjacent intervals for one-dimensional free fermions after applying the fermionic PT. We show that in general there is a freedom in the definition of such operation which leads to two different definitions of PT: the resulting density matrix is Hermitian in one case, while it becomes pseudo-Hermitian in the other case. Using the path-integral formalism, we analytically compute the leading order term of the moments in both cases and derive the distribution of the corresponding eigenvalues over the complex plane. We further verify our analytical findings by checking them against numerical lattice calculations.

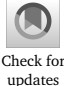
# 1  Introduction

Entanglement is an intrinsic property of quantum systems beyond classical physics. Having efficient frameworks to compute entanglement between two parts of a system is essential not only for fundamental interests such as characterizing phases of matter [1–4] and spacetime physics [5] but also for application purposes such as identifying useful resources to implement quantum computing processes. For a bipartite Hilbert space $\mathcal{H}_A \otimes \mathcal{H}_B$, it is easy to determine whether a pure state $|\Psi\rangle$ is entangled or not: a product state, i.e. any state of the form $|\Phi_A\rangle \otimes |\Phi_B\rangle$, is unentangled (separable), while a superposition state $|\Psi\rangle = \sum_i \alpha_i |\Phi_A^{(i)}\rangle \otimes |\Phi_B^{(i)}\rangle$, where $|\Phi_{A/B}^{(i)}\rangle$ is a set of local orthogonal states, is entangled. The amount of entanglement in a given state can be quantified by the entropy of information within either subsystem $A$ or $B$, in the form of the von Neumann entropy

$$S(\rho_A) = -\text{Tr}(\rho_A \ln \rho_A) = -\sum_i \alpha_i^2 \ln \alpha_i^2, \tag{1}$$

or the Rényi entanglement entropies (REEs)

$$\mathcal{R}_n(\rho_A) = \frac{1}{1-n} \ln \text{Tr}(\rho_A^n) = \frac{1}{1-n} \ln \sum_i \alpha_i^{2n}, \tag{2}$$

where $\rho_A = \text{Tr}_B(|\Psi\rangle \langle \Psi|) = \sum_i \alpha_i^2 |\Phi_A^{(i)}\rangle \langle \Phi_A^{(i)}|$ is the reduced density matrix acting on $\mathcal{H}_A$. Notice that $S(\rho_A) = S(\rho_B)$ and $\mathcal{R}_n(\rho_A) = \mathcal{R}_n(\rho_B)$ and clearly, $S, \mathcal{R}_n \geq 0$ where the equality holds for a product state. For analytical calculations, $S$ is usually obtained from $\mathcal{R}_n$ via $S = \lim_{n \to 1} \mathcal{R}_n$.

It is well-known that eigenvalues of density matrices, i.e. the entanglement spectrum, contains more information than merely the entanglement entropies. The entanglement spectrum has been studied and utilized toward better understanding of the phases of matter [6–23], broken-symmetry phases [24–28], and more exotic phases such as many-body localized states [29–31]. In particular, in the context of conformal field theories (CFTs) in (1+1)d

the distribution of eigenvalues was analytically derived [32] and was shown to obey a universal scaling function which depends only on the central charge of the underlying CFT. The obtained scaling function for the distribution of the entanglement spectrum at criticality was further substantiated numerically [33, 34], especially for matrix product state representation at critical points [35–37].

It turned out that extending the above ideas to mixed states where the system is described by a density matrix $\rho$ is not as easy as it may seem. A product state $\rho = \rho_A \otimes \rho_B$ is similarly unentangled. However, a large class of states, called separable states, in the form of $\rho = \sum_i p_i \rho_A^{(i)} \otimes \rho_B^{(i)}$ with $p_i \geq 0$ are classically correlated and do not contain any amount of entanglement. Hence, the fact that superposition implies entanglement in pure states does not simply generalize to the entanglement in mixed states. The positive partial transpose (PPT) [38–44] is a test designed to diagnose separable states based on the fact that density matrices are positive semi-definite operators. The PT of a density matrix $\rho = \sum_i \rho_{ijkl} |e_A^{(i)}, e_B^{(j)}\rangle \langle e_A^{(k)}, e_B^{(l)}|$ written in a local orthonormal basis $\{|e_A^{(k)}\rangle, |e_B^{(j)}\rangle\}$ is defined by exchanging the indices of subsystem $A$ (or $B$) as in

$$\rho^{T_A} = \sum_i \rho_{ijkl} |e_A^{(k)}, e_B^{(j)}\rangle \langle e_A^{(i)}, e_B^{(l)}|. \tag{3}$$

Note that $\rho^{T_A}$ is a Hermitian operator and the PPT test follows by checking whether or not $\rho^{T_A}$ contains any negative eigenvalue. A separable state passes the PPT test, i.e. all the eigenvalues of $\rho^{T_A}$ are non-negative, whereas an inseparable (i.e., entangled) state yields negative eigenvalues after PT[1]. Hence, the PPT criterion can be used to decide whether a given density matrix is separable or not. Similar to the entropic measures of pure-state entanglement in (1) and (2), the (logarithmic) entanglement negativity associated with the spectrum of the partially transposed density matrix is defined as a candidate to quantify mixed-state entanglement [47–49],

$$\mathcal{N}(\rho) = \frac{\left\|\rho^{T_A}\right\| - 1}{2}, \tag{4}$$

$$\mathcal{E}(\rho) = \ln\left\|\rho^{T_A}\right\|, \tag{5}$$

where $\|A\| = \text{Tr}\sqrt{AA^\dagger}$ is the trace norm. When $A$ is Hermitian, the trace norm is simplified into the sum of the absolute value of the eigenvalues of $A$. Hence, the above quantities measure the *negativity* of the eigenvalues of $\rho^{T_A}$. It is also useful to define the moments of the PT (aka Rényi negativity) via

$$\mathcal{N}_n(\rho) = \ln \text{Tr}(\rho^{T_A})^n, \tag{6}$$

where the logarithmic negativity is obtained from analytic continuation

$$\mathcal{E}(\rho) = \lim_{n \to 1/2} \mathcal{N}_{2n}(\rho). \tag{7}$$

Note that for a pure state $\rho = |\Psi\rangle\langle\Psi|$, we have $\mathcal{E}(\rho) = \mathcal{R}_{1/2}(\rho_A)$ where $\rho_A$ is the reduced density matrix on $\mathcal{H}_A$. The entanglement negativity has been used to characterize mixed states in various quantum systems such as in harmonic oscillator chains [50–58], quantum spin models [59–68], (1+1)d conformal and integrable field theories [69–76], topologically ordered phases of matter in (2+1)d [77–81], and in out-of-equilibrium situations [82–87], as well as

---

[1] A technical point is that there exists a set of inseparable states which also pass the PPT test [45]. They are said to contain bound entanglement which cannot be used for quantum computing processes such as teleportation [46]. This issue is beyond the scope of our paper and we do not elaborate further here.

holographic theories [88–93] and variational states [94–97]. Moreover, the PT was used to construct topological invariants for symmetry protected topological (SPT) phases protected by anti-unitary symmetries [98–101] and there are experimental proposals to measure it with cold atoms [102, 103].

Unlike the entanglement spectrum which has been studied extensively, less is known about the spectrum of partially transposed density matrices in many-body systems. It is true that the PPT test which predates the entanglement spectrum is based on the eigenvalues of the PT, but the test only uses the sign of the eigenvalues. Therefore, studying the spectrum of the PT could be useful in characterizing quantum phases of matter. Furthermore, the fact that PT is applicable to extract entanglement at finite-temperature states and that the eigenvalues have a sign structure (positive/negative) may help unravel some new features beyond the entanglement spectrum. Recently, the distribution of eigenvalues of the PT, dubbed as *the negativity spectrum*, was studied for CFTs in (1+1)d [72]. It was found that the negativity spectrum is universal and depends only on the central charge of the CFT, similar to the entanglement spectrum, while the precise form of the spectrum depends on the sign of the eigenvalues. This dependence is weak for bulk eigenvalues, whereas it is strong at the spectrum edges.

In this paper, we would like to study the negativity spectrum in fermionic systems. The PT of fermionic density matrices however involves some subtleties due to the Fermi statistics (i.e., anti-commutation relation of fermion operators). Initially, a procedure for the PT of fermions based on the fermion-boson mapping (Jordan-Wigner transformation) was proposed [104] and was also used in the subsequent studies [105–111]. However, this definition turned out to cause certain inconsistencies within fermionic theories such as violating the additivity property and missing some entanglement in topological superconductors, and give rise to incorrect classification of time-reversal symmetric topological insulators and superconductors. Additionally, according to this definition it is computationally hard to find the PT (and calculate the entanglement negativity) even for free fermions, since the PT of a fermionic Gaussian state is not Gaussian. This motivates us to use another way of implementing a fermionic PT which was proposed recently by some of us in the context of time-reversal symmetric SPT phases of fermions [100,101,112]. This definition does not suffer from the above issues and at the same time the associated entanglement quantity is an entanglement monotone [113]. From a practical standpoint, the latter definition has the merit that the partially transposed Gaussian state remains Gaussian and hence can be computed efficiently for free fermions. A detailed survey of differences between the two definitions of PT from both perspectives of quantum information and condensed matter theory (specifically, topological phases of fermions) is discussed in Refs. [112,113].

Before we get into details of the fermionic PT in the coming sections, let us finish this part with a summary of our main findings. We study the distribution of the many-body eigenvalues $\lambda_i$ of the partially transposed reduced density matrix,

$$P(\lambda) = \sum_i \delta(\lambda - \lambda_i) \tag{8}$$

for one-dimensional free fermions. As a lattice realization, we consider the hopping Hamiltonian on a chain

$$\hat{H} = -\sum_j [t(f_{j+1}^\dagger f_j + \text{H.c.}) + \mu f_j^\dagger f_j], \tag{9}$$

where the fermion operators $f_j$ and $f_j^\dagger$ obey the anti-commutation relation $\{f_i, f_j^\dagger\} = f_i f_j^\dagger + f_j^\dagger f_i = \delta_{ij}$ and $\{f_i, f_j\} = \{f_i^\dagger, f_j^\dagger\} = 0$.

Recall that using the regular (matrix) PT – we will refer to it as the *bosonic* PT –, which applies to generic systems where local operators commute, the obtained PT density matrix is a

Hermitian operator and its eigenvalues are either negative or positive. However, it turned out that for fermions a consistent definition of PT involves a phase factor as we exchange indices in (3) and in general one can define two types of PT operation. As we will explain in detail, these two types correspond to the freedom of spacetime boundary condition for fermions associated with the fermion-number parity symmetry. We reserve $\rho^{T_A}$ and $\rho^{\widetilde{T}_A}$ to denote the fermionic PT which leads to anti-periodic (untwisted) and periodic (twisted) boundary conditions along fundamental cycles of the spacetime manifold, respectively. We should note that $\rho^{T_A}$ is pseudo-Hermitian[2] and may contain complex eigenvalues, while $\rho^{\widetilde{T}_A}$ is Hermitian and its eigenvalues are real. We use the spacetime path integral formulation to analytically calculate the negativity spectrum. In the case of $\rho^{\widetilde{T}_A}$, we obtain results very similar to those of previous CFT work [72], where the distribution of positive and negative eigenvalues are described by two universal functions. In the case of $\rho^{T_A}$, we observe that the eigenvalues are complex but they have a pattern and fall on six branches in complex plane with a *quantized* complex phase of $\angle\lambda = 2\pi n/6$. We show that the spectrum is reflection symmetric with respect to the real axis and the eigenvalue distributions are described by four universal functions along $\angle\lambda = 0, \pm 2\pi/6, \pm 4\pi/6, \pi$ branches. We further verify our findings by checking them against numerical lattice simulations.

The rest of our paper is organized as follows: in Section 2 we provide a brief review of partial transpose for fermions, in Section 3 we discuss the spacetime path-integral formulation of the moments of partially transposed density matrices. The spectrum of the twisted and untwisted partial transpose is analytically derived in Section 4 for different geometries, where numerical checks with free fermions on the lattice are also provided. We close our discussion by some concluding remarks in Section 5. In several appendices, we give further details of the analytical calculations and make connections with other related concepts.

## 2 Preliminary remarks

In this section, we review some basic materials which we use in the next sections: the definition of PT for fermions, how to extract the distribution of the eigenvalues of an operator from its moments, and some properties of partially transposed Gaussian states.

### 2.1 Twisted and untwisted partial transpose for fermions

In this part, we briefly discuss some background materials on our definitions of PT for fermions. More details can be found in Refs. [112,113]. We consider a fermionic Fock space $\mathcal{H}$ generated by $N$ local fermionic modes $f_j$, $j = 1, \cdots, N$. The Hilbert space is spanned by $|n_1, n_2, \cdots, n_N\rangle$ which is a string of occupation numbers $n_j = 0, 1$. We define the Majorana (real) fermion operators in terms of canonical operators as

$$c_{2j-1} := f_j^\dagger + f_j, \quad c_{2j} := i(f_j - f_j^\dagger), \quad j = 1, \ldots, N. \tag{10}$$

These operators satisfy the commutation relation $\{c_j, c_k\} = 2\delta_{jk}$ and generate a Clifford algebra. Any operator $X$ acting on $\mathcal{H}$ can be expressed in terms of a polynomial of $c_j$'s,

$$X = \sum_{k=1}^{2N} \sum_{p_1 < p_2 \cdots < p_k} X_{p_1 \cdots p_k} c_{p_1} \cdots c_{p_k}, \tag{11}$$

---

[2]A pseudo-Hermitian operator $H$ is defined by $\eta H^\dagger \eta^{-1} = H$ with $\eta^2 = 1$ where $\eta$ is a unitary Hermitian operator satisfying $\eta^\dagger \eta = \eta \eta^\dagger = 1$ and $\eta = \eta^\dagger$. Essentially, pseudo-Hermiticity is a generalization of Hermiticity, in that it implies Hermiticity when $\eta = 1$.

where $X_{p_1 \cdots p_k}$ are complex numbers and fully antisymmetric under permutations of $\{1, \ldots, k\}$. A density matrix has an extra constraint, i.e., it commutes with the total fermion-number parity operator, $[\rho, (-1)^F] = 0$ where $F = \sum_j f_j^\dagger f_j$. This constraint entails that the Majorana operator expansion of $\rho$ only contains even number of Majorana operators, i.e., $k$ in the above expression is even.

To study the entanglement, we consider a bipartite Hilbert space $\mathcal{H}_A \otimes \mathcal{H}_B$ spanned by $f_j$ with $j = 1, \cdots, N_A$ in subsystem $A$ and $j = N_A + 1, \cdots, N_A + N_B$ in subsystem $B$. Then, a generic density matrix on $\mathcal{H}_A \otimes \mathcal{H}_B$ can be expanded in Majorana operators as

$$\rho = \sum_{k_1, k_2}^{k_1 + k_2 = \text{even}} \rho_{p_1 \cdots p_{k_1}, q_1 \cdots q_{k_2}} a_{p_1} \cdots a_{p_{k_1}} b_{q_1} \cdots b_{q_{k_2}}, \tag{12}$$

where $\{a_j\}$ and $\{b_j\}$ are Majorana operators acting on $\mathcal{H}^A$ and $\mathcal{H}^B$, respectively, and the even fermion-number parity condition is indicated by the condition $k_1 + k_2 = \text{even}$. Our definition of the PT for fermions is given by [101, 112]

$$\rho^{T_A} := \sum_{k_1, k_2}^{k_1 + k_2 = \text{even}} \rho_{p_1 \cdots p_{k_1}, q_1 \cdots q_{k_2}} i^{k_1} a_{p_1} \cdots a_{p_{k_1}} b_{q_1} \cdots b_{q_{k_2}}, \tag{13}$$

and similarly for $\rho^{T_B}$. It is easy to see that the subsequent application of the PT with respect to the two subsystems leads to the full transpose $(\rho^{T_A})^{T_B} = \rho^T$, i.e. reversing the order of Majorana fermion operators. In addition, the definition (13) implies that

$$(\rho^{T_A})^\dagger = (-1)^{F_A} \rho^{T_A} (-1)^{F_A}, \tag{14}$$

$$(\rho^{T_A})^{T_A} = (-1)^{F_A} \rho (-1)^{F_A}, \tag{15}$$

where $(-1)^{F_A}$ is the fermion-number parity operator on $H_A$, i.e. $F_A = \sum_{j \in A} f_j^\dagger f_j$. The first identity, namely the pseudo-Hermiticity, can be understood as a consequence of the fact that $(\rho^{T_A})^\dagger$ is defined the same as (13) by replacing $i^{k_1}$ with $(-i)^{k_1}$. The second identity reflects the fact that the fermionic PT is related to the action of time-reversal operator of spinless fermions in the Euclidean spacetime [101]. We should note that the matrix resulting from the PT is not necessarily Hermitian and may have complex eigenvalues, although $\text{Tr} \rho^{T_A} = 1$. The existence of complex eigenvalues is a crucial property which was used in the context of SPT invariants to show that the complex phase of $\text{Tr}(\rho \rho^{T_A})$, which represents a partition function on a non-orientable spacetime manifold, is a topological invariant. For instance, $\text{Tr}(\rho \rho^{T_A}) = e^{i2\pi\nu/8}$ for time-reversal symmetric topological superconductors (class BDI) which implies the $\mathbb{Z}_8$ classification. (Here $\nu \in \mathbb{Z}_8$ is the topological invariant). Nevertheless, we may still use Eq. (5) to define an analog of entanglement negativity for fermions and calculate the trace norm in terms of square root of the eigenvalues of the composite operator $\rho_\times = [(\rho^{T_A})^\dagger \rho^{T_A}]$, which is a Hermitian operator with real positive eigenvalues. On the other hand, from Eq. (14) we realize that $\rho_\times = (\rho^{\widetilde{T}_A})^2$ where we introduce the *twisted* PT by

$$\rho^{\widetilde{T}_A} := \rho^{T_A} (-1)^{F_A}. \tag{16}$$

It is easy to see from Eq. (14) that this operator is Hermitian and then similar to the bosonic PT always contains real eigenvalues. It is worth noting that

$$(\rho^{\widetilde{T}_A})^{\widetilde{T}_A} = \rho, \tag{17}$$

in contrast with the *untwisted* PT (15). As we will see shortly, this difference between $\rho^{T_A}$ and $\rho^{\widetilde{T}_A}$ in the operator formalism will show up as anti-periodic and periodic boundary conditions across the fundamental cycles of spacetime manifold in the path-integral formalism. The central result of our paper is to report analytical results for the spectrum of $\rho^{T_A}$ and $\rho^{\widetilde{T}_A}$.

## 2.2 The moment problem

In the replica approach to logarithmic negativity (5) and negativity spectrum, one first has to calculate the moments of PT, aka Rényi negativity (RN),

$$\mathcal{N}_n^{(\text{ns})}(\rho) = \ln \text{Tr}[(\rho^{T_A})^n], \qquad \mathcal{N}_n^{(\text{r})}(\rho) = \ln \text{Tr}[(\rho^{\tilde{T}_A})^n], \tag{18}$$

which are fermionic counterparts of the bosonic definition in Eq. (6). The superscripts (ns) and (r) stand for Neveu-Schwarz and Ramond respectively (the reason for this will be clear from the path integral representation of such quantities, see Section 3 below). Thus, the analog of analytic continuation (7) to obtain the logarithmic negativity is

$$\mathcal{E}(\rho) = \lim_{n \to 1/2} \mathcal{N}_{2n}^{(\text{r})}. \tag{19}$$

In the following, we review a general framework to analytically obtain the distribution of eigenvalues of density matrix (or its transpose) from the moments. This method was originally used to derive the entanglement spectrum of (1+1)d CFTs [32]. Suppose we have an operator $\mathcal{O}$ whose moments are of the form

$$R_n := \text{Tr}[\mathcal{O}^n]. \tag{20}$$

In terms of the eigenvalues of $\mathcal{O}$, $\{\lambda_j\}$, we have $R_n = \sum_j \lambda_j^n = \int P(\lambda) \lambda^n d\lambda$, where $P(\lambda)$ is the associated distribution function (see Eq. (8)). The goal is to find $P(\lambda)$ by making use of the specific form of $R_n$ in (20). The essential idea is to compute the Stieltjes transform

$$f(s) := \frac{1}{\pi} \sum_{n=1}^{\infty} R_n s^{-n} = \frac{1}{\pi} \int d\lambda \frac{\lambda P(\lambda)}{s - \lambda}. \tag{21}$$

Assuming that the eigenvalues are real, the distribution function can be easily read off from the relation

$$P(\lambda) = \frac{1}{\lambda} \lim_{\epsilon \to 0} \text{Im} f(\lambda - i\epsilon). \tag{22}$$

In the following we are going to focus on the complementary cumulative distribution function or simply the tail distribution, being a very simple object to be accessed for numerical comparison

$$n(\lambda) = \int_{\lambda}^{\lambda_{\max}} d\lambda P(\lambda). \tag{23}$$

For specific types of operators such as the density matrices and their PT in (1+1)d CFTs, the moments can be cast in the form,

$$R_n = r_n \exp\left(-bn + \frac{a}{n}\right), \qquad \forall n, \tag{24}$$

where $a, b \in \mathbb{R}$, $b > 0$ and $r_n$ are non-universal constant. In such cases, the distribution function is found to be [63]

$$P(\lambda; a, b) = \begin{cases} \frac{a\,\theta(e^{-b} - \lambda)}{\lambda \sqrt{a \ln(e^{-b}/\lambda)}} I_1(2\sqrt{a \ln(e^{-b}/\lambda)}) + \delta(e^{-b} - \lambda), & a > 0, \\ \frac{-|a|\,\theta(e^{-b} - \lambda)}{\lambda \sqrt{|a| \ln(e^{-b}/\lambda)}} J_1(2\sqrt{|a| \ln(e^{-b}/\lambda)}) + \delta(e^{-b} - \lambda), & a < 0, \end{cases} \tag{25}$$

and the corresponding tail distribution is given by

$$n(\lambda; a, b) = \begin{cases} I_0(2\sqrt{a \ln(e^{-b}/\lambda)}), & a > 0, \\ J_0(2\sqrt{|a| \ln(e^{-b}/\lambda)}), & a < 0, \end{cases} \tag{26}$$

where $J_\alpha(x)$ and $I_\alpha(x)$ are the regular Bessel functions and modified Bessel functions of the first kind, respectively. Note that (25) and (26) are derived by ignoring the presence of the constants $r_n$ in (24). This relies on the assumption that they do not change significantly upon varying $n$, i.e., $\lim_{n\to\infty} \frac{1}{n} \ln r_n < \infty$. The very same assumption has been adopted for the entanglement and bosonic negativity spectrum in Refs. [32] and [72] where the derived distribution functions agree with the numerically obtained spectra.

### 2.3 Partial transpose of Gaussian states

Here, we discuss how to compute the spectrum of the PT of a Gaussian state from the corresponding covariance matrix. The idea is similar to that of the entanglement spectrum, while there are some differences as the covariance matrix associated with the partially transposed density matrix may contain complex eigenvalues. Before we continue, let us summarize the structure of the many-body spectrum of $\rho^{T_A}$ and $\rho^{\widetilde{T_A}}$ for free fermions,

$$\text{Spec}[\rho^{T_A}] : \begin{cases} (\lambda_i, \lambda_i^*) & \text{Im}[\lambda_i] \neq 0, \\ (\lambda_i, \lambda_i) & \text{Im}[\lambda_i] = 0, \quad \lambda_i < 0, \\ \lambda_i & \text{Im}[\lambda_i] = 0, \quad \lambda_i > 0, \end{cases} \tag{27}$$

$$\text{Spec}[\rho^{\widetilde{T_A}}] : \begin{cases} (\lambda_i, \lambda_i) & \lambda_i < 0, \\ \lambda_i & \lambda_i > 0, \end{cases} \tag{28}$$

where repeating values mean two fold degeneracy. We should note that the pseudo-Hermiticity of $\rho^{T_A}$ (14) ensures that the complex-valued subset of many-body eigenvalues of $\rho^{T_A}$ appear in complex conjugate pairs. This property is general and applicable to any density matrix beyond free fermions. An immediate consequence of this property is that any moment of $\rho^{T_A}$ is guaranteed to be real-valued.

A Gaussian density matrix in the Majorana fermion basis (10) is defined by

$$\rho_\Omega = \frac{1}{\mathcal{Z}(\Omega)} \exp\left( \frac{1}{4} \sum_{j,k=1}^{2N} \Omega_{jk} c_j c_k \right), \tag{29}$$

where $\Omega$ is a pure imaginary antisymmetric matrix and $\mathcal{Z}(\Omega) = \pm\sqrt{\det\left(2\cosh\frac{\Omega}{2}\right)}$ is the normalization constant. We should note that the spectrum of $\Omega$ is in the form of $\pm\omega_j$, $j = 1, \ldots, N$ and the $\pm$ sign ambiguity in $\mathcal{Z}(\Omega)$ is related to the square root of determinant where we need to choose one eigenvalue for every pair $\pm\omega_j$. The sign is fixed by the Pfaffian. This density matrix can be uniquely characterized by its covariance matrix,

$$\Gamma_{jk} = \frac{1}{2}\text{Tr}(\rho_\Omega[c_j, c_k]), \tag{30}$$

which is a $2N \times 2N$ matrix. These two matrices are related by

$$\Gamma = \tanh\left(\frac{\Omega}{2}\right), \qquad e^\Omega = \frac{\mathbb{I} + \Gamma}{\mathbb{I} - \Gamma}. \tag{31}$$

Furthermore, one can consider a generic Gaussian operator which is also defined through Eq. (29), but without requiring that the spectrum is pure imaginary. An equivalent description in terms of the covariance matrix is also applicable for such operators. The only difference is that the eigenvalues do not need to be real. Let us recall how Rényi entropies (2) are computed for Gaussian states. The density matrix (29) can be brought into a diagonal form $\rho_\Omega = \mathcal{Z}^{-1} \exp\left(\frac{i}{2} \sum_n \omega_n d_{2n} d_{2n-1}\right)$, where $\omega_n$ is obtained from an orthogonal transformation

of $\Omega$. In terms of the eigenvalues of $\Gamma$, denoted by $\pm \nu_j$, we have $\rho_\Omega = \prod_n (1 + i \nu_n d_{2n} d_{2n-1})/2$, leading to

$$\mathcal{R}_n(\rho) = \frac{1}{1-n} \sum_{j=1}^{N} \ln\left[\left(\frac{1-\nu_j}{2}\right)^n + \left(\frac{1+\nu_j}{2}\right)^n\right]. \tag{32}$$

We consider a density matrix on a bipartite Hilbert space (12) where the covariance matrix takes a block matrix form as

$$\Gamma = \begin{pmatrix} \Gamma_{AA} & \Gamma_{AB} \\ \Gamma_{BA} & \Gamma_{BB} \end{pmatrix}. \tag{33}$$

Here, $\Gamma_{AA}$ and $\Gamma_{BB}$ denote the reduced covariance matrices of subsystems $A$ and $B$, respectively; while $\Gamma_{AB} = \Gamma_{BA}^\dagger$ describes the correlations between them. We define the covariance matrix associated with a partially transposed Gaussian state by

$$\Gamma_\pm = \begin{pmatrix} -\Gamma_{AA} & \pm i\Gamma_{AB} \\ \pm i\Gamma_{BA} & \Gamma_{BB} \end{pmatrix}, \tag{34}$$

where $[\Gamma_+]_{ij} = \frac{1}{2}\text{Tr}(\rho^{T_A}[c_i, c_j])$ and $[\Gamma_-]_{ij} = \frac{1}{2}\text{Tr}(\rho^{T_A \dagger}[c_i, c_j])$. We should note that $\Gamma_+$ and $\Gamma_-$ have identical eigenvalues while they do not necessarily commute $[\Gamma_+, \Gamma_-] \neq 0$. In general, the eigenvalues of $\Gamma_+$ appear in quartets $(\pm \nu_k, \pm \nu_k^*)$ when $\text{Re}[\nu_k] \neq 0$ and $\text{Im}[\nu_k] \neq 0$ or doublet $\pm \nu_k$ when $\text{Re}[\nu_k] = 0$ (i.e., pure imaginary) or $\text{Im}[\nu_k] = 0$ (i.e., real) . $\pm$ is because of skew symmetry $\Gamma_\pm^T = -\Gamma_\pm$. In addition, the pseudo-Hermiticity of $\rho^{T_A}$ (14) implies that

$$\Gamma_\pm^\dagger = U_1 \Gamma_\pm U_1, \tag{35}$$

where $U_1 = (-\mathbb{I}_A \oplus \mathbb{I}_B)$ is the matrix associated with the operator $(-1)^{F_A}$. This means that for every eigenvalue $\nu_k$ its complex conjugate $\nu_k^*$ is also an eigenvalue. As a result, the moments of PT can be written as

$$\mathcal{N}_n^{(\text{ns})} = \sum_{j=1}^{N} \ln\left|\left(\frac{1-\nu_j}{2}\right)^n + \left(\frac{1+\nu_j}{2}\right)^n\right|. \tag{36}$$

Note that the sum is now over half of the eigenvalues (say in the upper half complex plane), due to the structure discussed above.

For $\rho^{\widetilde{T_A}}$ we use the multiplication rule for the Gaussian operators where the resulting Gaussian matrix is given by

$$e^{\widetilde{\Omega}_\pm} = \frac{\mathbb{I} + \Gamma_\pm}{\mathbb{I} - \Gamma_\pm} U_1, \tag{37}$$

which is manifestly Hermitian due to the identity (35). Using Eq. (29), the normalization factor is found to be $\mathcal{Z}_{\widetilde{T_A}} = \text{Tr}(\rho^{\widetilde{T_A}}) = \text{Tr}[\rho(-1)^{F_A}] = \sqrt{\det \Gamma_{AA}}$. From (31) we construct the covariance matrix $\widetilde{\Gamma}_\pm = \tanh(\widetilde{\Omega}/2)$ and compute the moments of $\rho^{\widetilde{T_A}}$ by

$$\mathcal{N}_n^{(\text{r})} = \sum_{j=1}^{N} \ln\left|\left(\frac{1-\tilde{\nu}_j}{2}\right)^n + \left(\frac{1+\tilde{\nu}_j}{2}\right)^n\right| + n \ln \mathcal{Z}_{\widetilde{T_A}}, \tag{38}$$

where $\pm \tilde{\nu}_j$ are eigenvalues of $\widetilde{\Gamma}_\pm$ which are guaranteed to be real. Consequently, the logarithmic negativity (19) is given by

$$\mathcal{E} = \sum_{j=1}^{N} \ln\left[\left|\frac{1-\tilde{\nu}_j}{2}\right| + \left|\frac{1+\tilde{\nu}_j}{2}\right|\right] + \ln \mathcal{Z}_{\widetilde{T_A}}. \tag{39}$$

For particle-number conserving systems such as the lattice model in (9), the covariance matrix is simplified into the form $\Gamma = \sigma_2 \otimes \gamma$ where $\gamma = (\mathbb{I} - 2C)$ and $C_{ij} = \text{Tr}(\rho f_i^\dagger f_j)$ is the correlation matrix and $\sigma_2$ is the second Pauli matrix acting on the even/odd indices of Majorana operators $(c_{2j}, c_{2j-1})$. In this case, the transformed correlation matrix for $\rho^{T_A}$ is given by

$$\gamma_\pm = \begin{pmatrix} -\gamma_{AA} & \pm i \gamma_{AB} \\ \pm i \gamma_{BA} & \gamma_{BB} \end{pmatrix}. \tag{40}$$

The eigenvalues can be divided to two categories: complex eigenvalues $v_k$, $\text{Im}[v_k] \neq 0$ and real eigenvalues $u_k$, $\text{Im}[u_k] = 0$. The pseudo-Hermiticity property leads to the identity $\gamma_\pm^\dagger = U_1 \gamma_\pm U_1$ which implies that complex eigenvalues appear in pairs $(v_k, v_k^*)$. Therefore, the many-body eigenvalues follow the form,

$$\lambda_{\sigma, \sigma'} = \prod_{\sigma_l} \frac{1 + \sigma_l u_l}{2} \prod_{\sigma_k = \sigma_k'} \frac{1 + |v_k|^2 + 2\sigma_k \text{Re}[v_k]}{4} \prod_{\sigma_k = -\sigma_k'} \frac{1 - |v_k|^2 + 2\sigma_k i \text{Im}[v_k]}{4}, \tag{41}$$

where $\boldsymbol{\sigma} = \{\sigma_k = \pm\}$ is a string of signs. Clearly, the many-body eigenvalues appear in two categories as well: complex conjugate pairs $(\lambda_j, \lambda_j^*)$ and real eigenvalues which are not necessarily degenerate.

We can also derive a simple expression for the correlation matrix $\widetilde{C} = (\mathbb{I} - \widetilde{\gamma})/2$ associated with $\rho^{\widetilde{T}_A}$,

$$\widetilde{\gamma} = \begin{pmatrix} -\gamma_{AA}^{-1}(\mathbb{I}_A + \gamma_{AB}\gamma_{BA}) & i\gamma_{AA}^{-1}\gamma_{AB}\gamma_{BB} \\ i\gamma_{BA} & \gamma_{BB} \end{pmatrix}. \tag{42}$$

# 3 Spacetime picture for the moments of partial transpose

In the following two sections, we compute the moments of the partially transposed density matrix and ultimately the logarithmic negativity. First, we develop a general method using the replica approach [69, 70, 110] and provide an equivalent spacetime picture of the Rényi negativity.

Before we proceed, let us briefly review the replica approach to find the entanglement entropy. Next, we make connections to our construction of PT. A generic density matrix can be represented in the fermionic coherent state as

$$\rho = \int d\alpha d\bar{\alpha} \, d\beta d\bar{\beta} \, \rho(\bar{\alpha}, \beta) |\alpha\rangle \langle \bar{\beta}| e^{-\bar{\alpha}\alpha - \bar{\beta}\beta}, \tag{43}$$

where $\alpha$, $\bar{\alpha}$, $\beta$ and $\bar{\beta}$ are independent Grassmann variables and we omit the real-space (and possibly other) indices for simplicity. The trace formula then reads

$$Z_{\mathcal{R}_n} = \text{Tr}[\rho^n] = \int \prod_{i=1}^n d\psi_i d\bar{\psi}_i \prod_{i=1}^n \left[ \rho(\bar{\psi}_i, \psi_i) \right] e^{\sum_{i,j} \bar{\psi}_i T_{ij} \psi_j}, \tag{44}$$

where the subscripts in $\psi_i$ and $\bar{\psi}_i$ denote the replica indices and $T$ is called the twist matrix,

$$T = \begin{pmatrix} 0 & -1 & 0 & \dots \\ 0 & 0 & -1 & 0 \\ \vdots & \vdots & \ddots & -1 \\ 1 & 0 & \cdots & 0 \end{pmatrix}. \tag{45}$$

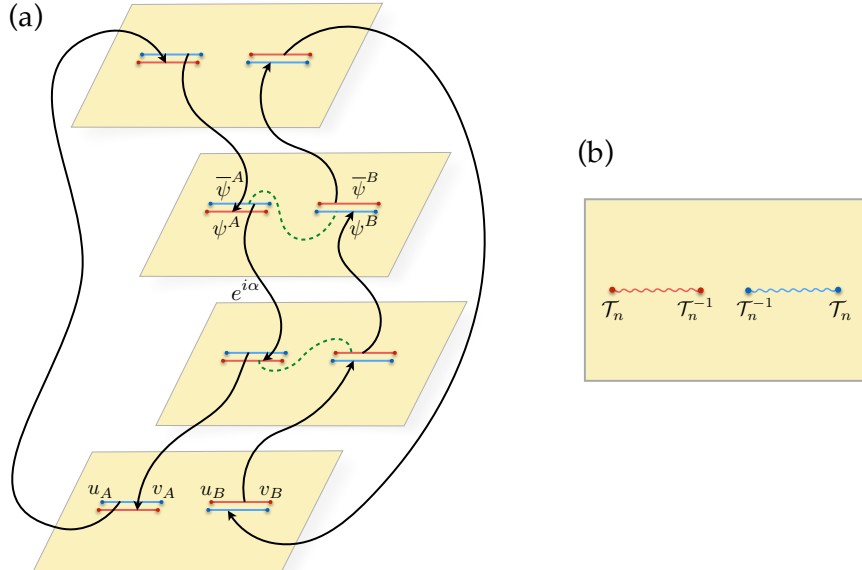

Figure 1: (a) Spacetime manifold associated with $Z_{\mathcal{N}_n}(\alpha)$, Eq. (53), for $n = 4$. The operator $e^{i\alpha F_A}$ twists the boundary condition of the cycles between two successive sheets, shown as the green path with dashed lines. (b) Equivalent picture in terms of twist field where we define a multi-component field on a single spacetime sheet.

The above expression can be viewed as a partition function on a $n$-sheet spacetime manifold where the $n$ flavors (replicas) $\psi_i$ are glued in order along the cuts. Alternatively, one can consider a multi-component field $\Psi = (\psi_1, \cdots, \psi_n)^T$ on a single-sheet spacetime. This way when we traverse a close path through the interval the field gets transformed as $\Psi \mapsto T\Psi$. Hence, each interval can be represented by two branch points $\mathcal{T}_n$ and $\mathcal{T}_n^{-1}$ –the so-called twist fields– and the REE of one interval can be written as a two-point correlator [114],

$$Z_{\mathcal{R}_n} = \langle \mathcal{T}_n(u) \mathcal{T}_n^{-1}(v) \rangle, \qquad (46)$$

where $u$ and $v$ denote the real space coordinates of the two ends of the interval defining the subsystem $A$.

Let us now derive analogous relations for the moments of partially transposed density matrix. Using the definition of the PT in the coherent state basis [112]

$$(|\psi_A, \psi_B\rangle \langle \bar{\psi}_A, \bar{\psi}_B|)^{T_A} = |i\bar{\psi}_A, \psi_B\rangle \langle i\psi_A, \bar{\psi}_B|, \qquad (47)$$

we write the general expression for the moments of $\rho^{T_A}$ as

$$Z_{\mathcal{N}_n}^{(\mathrm{ns})} = \mathrm{Tr}[(\rho^{T_A})^n] = \int \prod_{i=1}^n d\psi_i d\bar{\psi}_i \prod_{i=1}^n \left[ \rho(\bar{\psi}_i, \psi_i) \right] e^{\sum_{i,j} \bar{\psi}_{iA}[T^{-1}]_{ij}\psi_{jA}} e^{\sum_{i,j} \bar{\psi}_{iB} T_{ij}\psi_{jB}}, \qquad (48)$$

where $\psi_{js}$ and $\bar{\psi}_{js}$ refer to the field defined within the $s = A, B$ interval of $j$th replica. Here, we are dealing with two intervals where the twist matrices are $T$ and $T^{-1}$ as shown in Fig. 1(a). Therefore, it can be written as a four-point correlator (1(b))

$$Z_{\mathcal{N}_n}^{(\mathrm{ns})} = \langle \mathcal{T}_n^{-1}(u_A) \mathcal{T}_n(v_A) \mathcal{T}_n(u_B) \mathcal{T}_n^{-1}(v_B) \rangle. \qquad (49)$$

Note that the order of twist fields are reversed for the first interval.

From the coherent state representation, we can also write the moments of $\rho^{\widetilde{T}_A}$

$$Z_{\mathcal{N}_n}^{(\mathrm{r})} = \mathrm{Tr}[(\rho^{\widetilde{T}_A})^n] = \int \prod_{i=1}^{n} d\psi_i d\bar{\psi}_i \prod_{i=1}^{n} [\rho(\bar{\psi}_i, \psi_i)] e^{\sum_{i,j} \bar{\psi}_i^A \widetilde{T}_{ij} \psi_j^A} e^{\sum_{i,j} \bar{\psi}_i^B T_{ij} \psi_j^B}. \tag{50}$$

The twist matrix for interval A is modified to be

$$\widetilde{T} = \begin{pmatrix} 0 & \cdots & 0 & -1 \\ 1 & \ddots & \vdots & \vdots \\ 0 & 1 & 0 & 0 \\ \cdots & 0 & 1 & 0 \end{pmatrix}, \tag{51}$$

which can be viewed as a gauge transformed twist matrix $T^{-1}$. Analogously, Eq. (50) can be written in terms of a four-point correlator

$$Z_{\mathcal{N}_n}^{(\mathrm{r})} = \langle \widetilde{\mathcal{T}}_n^{-1}(u_A) \widetilde{\mathcal{T}}_n(v_A) \mathcal{T}_n(u_B) \mathcal{T}_n^{-1}(v_B) \rangle, \tag{52}$$

where $\widetilde{\mathcal{T}}_n$ and $\widetilde{\mathcal{T}}_n^{-1}$ are twist fields associated with $\widetilde{T}$.

For fermions with a global $U(1)$ gauge symmetry (i.e., particle-number conserving systems) there is a freedom to twist boundary condition along the fundamental cycles (e.g. the dashed-line path in Fig. 1(a)) of the spacetime manifold by a $U(1)$ phase (or holonomy). The boundary conditions are independent and in principle can be different for different pairs of sheets. If we assume a replica symmetry (i.e. uniform boundary conditions) $\psi_i \mapsto e^{i\alpha}\psi_i$, the expression for the PT moments in the operator formalism is given by

$$Z_{\mathcal{N}_n}(\alpha) = \mathrm{Tr}[(\rho^{T_A} e^{i\alpha F_A})^n]. \tag{53}$$

Let us mention that some related quantities such as $\mathrm{Tr}[(\rho \, e^{i\alpha F})^n]$ were previously introduced and dubbed charged entanglement entropies [115]. They were further used to determine symmetry resolved entanglement entropies which is the contribution from the density matrix to the entanglement entropies when projected onto a given particle-number sector [116,117].

From (53), we get a family of RN parametrized by $\alpha$. However, for a generic fermionic system (including superconductors), the $U(1)$ symmetry is reduced to $\mathbb{Z}_2$ fermion-parity symmetry. Hence, the two quantities of general interest would be

$$Z_{\mathcal{N}_n}(\alpha = \pi) = Z_{\mathcal{N}_n}^{(\mathrm{r})} = \mathrm{Tr}[(\rho^{\widetilde{T}_A})^n], \tag{54}$$

$$Z_{\mathcal{N}_n}(\alpha = 0) = Z_{\mathcal{N}_n}^{(\mathrm{ns})} = \mathrm{Tr}[(\rho^{T_A})^n]. \tag{55}$$

We should reemphasize that either quantities are described by a partition function on the same spacetime manifold (Fig. 1) as in the case of bosonic systems [70], while they differ in the boundary conditions for fundamental cycles of the manifold. In other words, $Z_{\mathcal{N}_n}^{(\mathrm{ns})}$ and $Z_{\mathcal{N}_n}^{(\mathrm{r})}$ correspond to anti-periodic (i.e., Neveu-Schwarz in CFT language) and periodic (Ramond) boundary conditions, respectively. This can be readily seen by comparing $T^{-1}$ and $\widetilde{T}$. These boundary conditions correspond to two replica-symmetric spin structures for the spacetime manifold. This is different from bosonic PT of fermionic systems [107, 110], where RN is given by sum over all possible spin structures. Essentially, the RNs associated with the two types of fermionic PT are identical to two terms in the expansion of bosonic PT in Ref. [107].

In what follows, we compute the two RNs for two partitioning schemes:

- Two **adjacent intervals** which is obtained by fusing the fields in $v_A$ and $u_B$. Hence, the RNs are given in terms of three-point correlators

$$Z_{\mathcal{N}_n}^{(\mathrm{ns})} = \langle \mathcal{T}_n^{-1}(u_A) \mathcal{T}_n^2(v_A) \mathcal{T}_n^{-1}(v_B) \rangle, \tag{56}$$

and

$$Z_{\mathcal{N}_n}^{(r)} = \langle \widetilde{\mathcal{T}}_n^{-1}(u_A)\mathcal{Q}_n^2(v_A)\mathcal{T}_n^{-1}(v_B)\rangle\,, \tag{57}$$

where we introduce the fusion of unlike twist fields,

$$\mathcal{Q}_n^2 := \mathcal{T}_n\widetilde{\mathcal{T}}_n. \tag{58}$$

- **Bipartite geometry** where the two intervals together form the entire system which is in the ground state. This time the RNs are obtained by further fusing the fields in $u_A$ and $v_B$ and the final expressions are therefore given by the two-point correlators

$$Z_{\mathcal{N}_n}^{(ns)} = \langle \mathcal{T}_n^{-2}(u_A)\mathcal{T}_n^2(v_A)\rangle\,, \tag{59}$$

and

$$Z_{\mathcal{N}_n}^{(r)} = \langle \mathcal{Q}_n^{-2}(u_A)\mathcal{Q}_n^2(v_A)\rangle\,. \tag{60}$$

## 4 The spectrum of partial transpose

As mentioned, the first step to compute the tail distribution of the eigenvalues of partially transposed density matrix is to find its moments. To this end, it is more convenient to work in a new basis where the twist matrices are diagonal and decompose the partition function of multi-component field $\Psi$ to $n$ decoupled partition functions. For REE, this leads to $Z_{\mathcal{R}_n} = \prod_{k=-(n-1)/2}^{(n-1)/2} Z_{k,n}$, where

$$Z_{k,n} = \langle \mathcal{T}_{k,n}(u)\mathcal{T}_{k,n}^{-1}(v)\rangle\,. \tag{61}$$

The monodromy condition for the field around $\mathcal{T}_{k,n}$ and $\mathcal{T}_{k,n}^{-1}$ are given by $\psi_k \mapsto e^{\pm i2\pi k/n}\psi_k$. The calculation of the above partition function can be further simplified in terms of correlators of vertex operators using the bosonization technique in (1+1)d. For instance, in the case of REE, (61) can be evaluated by [114]

$$Z_{k,n} = \langle V_k(u)V_{-k}(v)\rangle\,, \tag{62}$$

where $V_k(x) = e^{-i\frac{k}{n}\phi(x)}$ is the vertex operator and the expectation values is understood on the ground state of the scalar-field theory $\mathcal{L}_\phi = \frac{1}{8\pi}\partial_\mu\phi\partial^\mu\phi$. The correlation function of the vertex operators is found by

$$\langle V_{e_1}(z_1)\cdots V_{e_N}(z_N)\rangle \propto \prod_{i<j}\left|z_j - z_i\right|^{2e_ie_j}\,, \tag{63}$$

where $V_e(z) = e^{ie\phi(z)}$ is the vertex operator and $\sum_j e_j = 0$. Hence, we can write for the partition function

$$Z_{\mathcal{R}_n} \propto |u-v|^{-2\sum_k \frac{k^2}{n^2}}\,, \tag{64}$$

leading to the familiar result

$$\mathcal{R}_n = \frac{n+1}{6n}\ln|u-v| + \cdots \tag{65}$$

for the REE of 1d free fermions. Note that ellipses come from the proportionality constant in (64) which show sub-leading terms and may depend on microscopic details. In what follows, we apply the bosonization technique to evaluate $Z_{\mathcal{N}_n}^{(\text{ns})}$ and $Z_{\mathcal{N}_n}^{(\text{r})}$ similar to what we did for the REE. The scaling behavior of RNs in the lattice model is compared with the analytically predicted values of slopes (derived below) for various exponents $n = 1, \cdots, 7$ in Fig. 2, where the agreement is evident. We should note that the slope does not depend on the chemical potential $\mu$ in the Hamiltonian (9).

## 4.1 Spectrum of $\rho^{T_A}$

In the case of RN (48), we can carry out a similar *momentum* decomposition as

$$Z_{\mathcal{N}_n}^{(\text{ns})} = \prod_{k=-(n-1)/2}^{(n-1)/2} Z_{k,n}^{(\text{ns})}, \tag{66}$$

where

$$Z_{k,n}^{(\text{ns})} = \langle \mathcal{T}_{k,n}^{-1}(u_A)\mathcal{T}_{k,n}(v_A)\mathcal{T}_{k,n}(u_B)\mathcal{T}_{k,n}^{-1}(v_B)\rangle \tag{67}$$

is the partition function in the presence of four twist fields. We then use (63) to compute the above correlator for various subsystem geometries. We should note that the following results only include the leading order term in the scaling limit, $\ell_1, \ell_2 \to \infty$, where $\ell_1$ and $\ell_2$ are the length of $A$ and $B$ subsystems, respectively.

### 4.1.1 Adjacent intervals

Here, we consider adjacent intervals (c.f. upper panel of Fig. 2(a)). The final result is given by

$$Z_{k,n}^{(\text{ns})} = \begin{cases} \ell_1^{-4\frac{k^2}{n^2}}\ell_2^{-4\frac{k^2}{n^2}}(\ell_1+\ell_2)^{2\frac{k^2}{n^2}} & |k/n| < 1/3, \\ f(\ell_1,\ell_2;|k/n|)\cdot(\ell_1+\ell_2)^{2|\frac{k}{n}|(|\frac{k}{n}|-1)} & |k/n| > 1/3, \end{cases} \tag{68}$$

where $f(x, y; q) = \frac{1}{2}\left[x^{2(q-1)(-2q+1)}y^{2q(-2q+1)} + x \leftrightarrow y\right]$. Notice that the exponents change discontinuously as a function of $k$. This can be understood as a consequence of the $2\pi$ ambiguity of the $U(1)$ phase that the Fermi field acquires as it goes around the twist fields. Essentially, we need to find the dominant term with the lowest scaling dimension in the mode expansion (see Appendix A for more details). Adding up the terms in the $Z_{\mathcal{N}_n}^{(\text{ns})}$ expansion, the final expression in the limit of two equal-length intervals $\ell_1 = \ell_2$ is simplified into $\mathcal{N}_n^{(\text{ns})} = c_n \ln \ell + \cdots$ where

$$c_n = \begin{cases} -\frac{1}{3}\left(n - \frac{3}{2n}\right) & n = 6N, \\ -\frac{1}{3}\left(n - \frac{1}{n}\right) & n = 6N+1, 6N+5, \\ -\frac{1}{3}\left(n + \frac{1}{2n}\right) & n = 6N+2, 6N+4, \\ -\frac{1}{3}\left(n + \frac{3}{n}\right) & n = 6N+3, \end{cases} \tag{69}$$

where $N$ is a non-negative integer. It is worth recalling that for the bosonic systems, the spectrum of PT contains only positive and negative eigenvalues. As a result, we see even/odd effect for the moments. Here, however, the moments $Z_n^{(\text{ns})}$ have a cyclic behavior with a periodicity of six, which signals the possibility for the eigenvalues to appear with a multiple of $2\pi/6$ complex phase. As we will see below, this is indeed the case in our numerical calculations. We should also note that the above result can be obtained from the adjacent limit $v_A \to u_B$ of two disjoint intervals (67) as explained in Appendix B. Taking this limit is a bit tricky and was

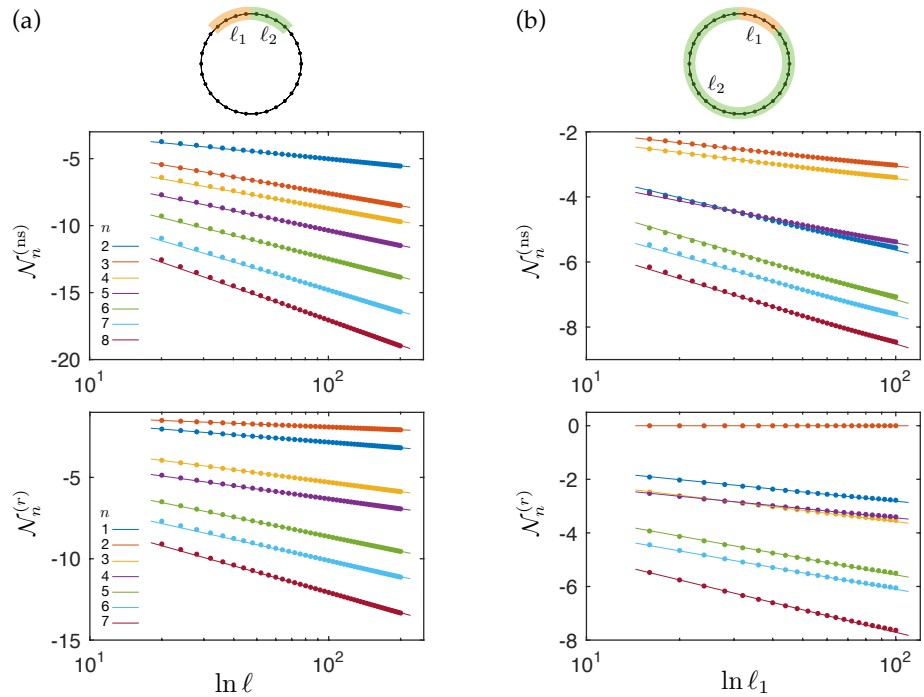

Figure 2: Comparison of numerical (dots) and analytical (solid lines) results for the scaling behavior of the moments of partial transpose (48) in the up row and (50) in the down row for two subsystem geometries: (a) two adjacent intervals, and (b) bipartite geometry. In (a), intervals have equal lengths $\ell_1 = \ell_2 = \ell$ and $20 \leq \ell \leq 200$ on an infinite chain. In (b), the total system size is $L = 400$ and $20 \leq \ell \leq 100$. Different colors correspond to different moments $n$.

previously overlooked in Ref. [110], where it was incorrectly deduced that $Z_{\mathcal{N}_n}^{(\mathrm{ns})} = 0$ for two adjacent intervals.

We now discuss the spectrum of $\rho^{T_A}$ for two adjacent intervals. It is instructive to look at the many-body eigenvalues as obtained in (41) from the single-body eigenvalues of the covariance matrix (40). From the numerical observation that $\mathrm{Im}(\nu_k) \neq 0$, we may drop the $u_l$ factor in (41). Hence, the many-body spectrum simplifies to

$$\lambda_{\boldsymbol{\sigma},\boldsymbol{\sigma}'} = \prod_{\sigma_k = \sigma'_k} \omega_{R\sigma_k} \prod_{\sigma_k = -\sigma'_k} \omega_{I\sigma'_k}, \tag{70}$$

where

$$\omega_{R\sigma_k} = \frac{1 + |\nu_k|^2 + 2\sigma_k \mathrm{Re}[\nu_k]}{4}, \tag{71a}$$

$$\omega_{I\sigma_k} = \frac{1 - |\nu_k|^2 + 2\sigma_k i \mathrm{Im}[\nu_k]}{4}, \tag{71b}$$

and $\sigma_k = \pm$ is a sign factor. We should note that the complex and negative real eigenvalues come from product of $\omega_{I\sigma_k}$. This fact immediately implies that for every complex eigenvalue $\lambda_j$, $\lambda_j^*$ is also in the spectrum, since $\omega_{I-\sigma_k} = \omega_{I\sigma_k}^*$. Moreover, the negative eigenvalues are at least two-fold degenerate.

In the case of free fermions, we numerically observe that $\omega_{I\pm} \to |\omega_{I\pm}| e^{\pm i \frac{2\pi}{6}}$ as we go towards the thermodynamic limit $N_A = N_B \to \infty$. As a result, the many-body eigenvalues are divided into two groups: first, real positive eigenvalues, and second, the complex or negative

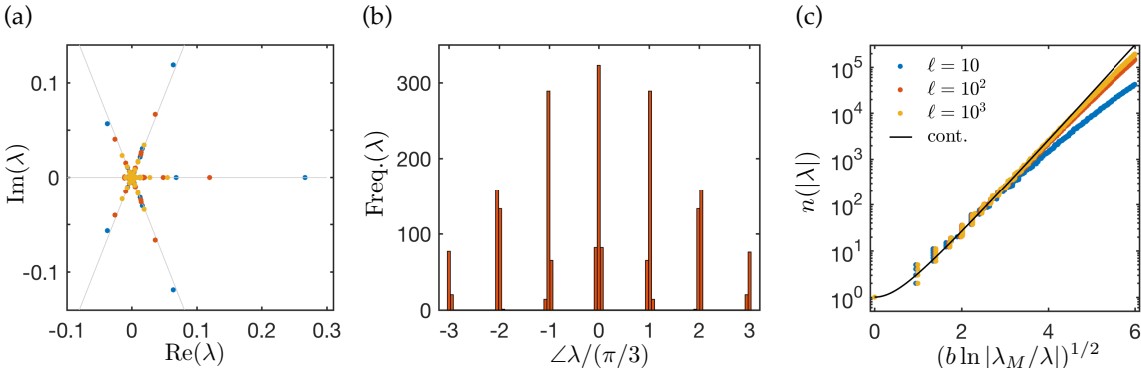

Figure 3: Spectral properties of $\rho^{T_A}$ for two adjacent intervals with length $\ell$ on an infinite chain. (a) Many-body eigenvalues are plotted over the complex plane. The solid gray lines are guides for the eyes and a hint for the phase quantization. (b) Histogram of complex phases of eigenvalues which indicates nearly quantized phases in units of $\pi/3$. (c) Tail distribution function of modulus of eigenvalues. The solid line is the analytical result (73b). To compute the many-body spectrum, we truncate the single-particle spectrum with the first 28 largest (in euclidean distance from $\pm 1$ on the complex plane) eigenvalues.

eigenvalues which take a regular form $\lambda_j \approx |\lambda_j| e^{\pm i \frac{\pi}{3} s_j}$ where $s_j = 1, 2, 3$. Figure 3(a) shows the numerical spectrum of $\rho^{T_A}$. To explicitly demonstrate the quantization of the complex phase of eigenvalues, we plot a histogram of the complex phase in Fig. 3(b) where sharp peaks at integer multiples of $\pi/3$ are evident. Due to this special structure of the eigenvalues, the moments of $\rho^{T_A}$ can be written as

$$
\begin{aligned}
Z_{\mathcal{N}_n}^{(\mathrm{ns})} &= \sum_k |\lambda_k|^n e^{\frac{i\pi n s_k}{3}} \\
&= \sum_j \lambda_{0j}^n + 2\cos\left(\frac{\pi n}{3}\right)\sum_j |\lambda_{1j}|^n + 2\cos\left(\frac{2\pi n}{3}\right)\sum_j |\lambda_{2j}|^n + \cos(n\pi)\sum_j |\lambda_{3j}|^n, \quad (72)
\end{aligned}
$$

where $\{\lambda_{\alpha j}\}, \alpha = 0, 1, 2, 3$ denote the eigenvalues along $\angle\lambda = \alpha\pi/3$ branches. Note that $\{\lambda_{0j}\}$, $\{\lambda_{3j}\}$, i.e., positive and negative real eigenvalues, are treated separately, while $\{\lambda_{1j}\}$ and $\{\lambda_{2j}\}$ represent the eigenvalues for both $\angle\lambda = \pm\pi/3$ and $\angle\lambda = \pm 2\pi/3$ branches. A consequence of Eq. (72) is that there are four linearly independent combinations of the eigenvalues in $Z_{\mathcal{N}_n}^{(\mathrm{ns})}$. This exactly matches the four possible scaling behaviors of $Z_{\mathcal{N}_n}^{(\mathrm{ns})}$ from our continuum field theory calculations (69).

As a first characterization of the negativity spectrum, we compute the distribution of modulus of eigenvalues. To this end, it is sufficient to consider $Z_{\mathcal{N}_n}^{(\mathrm{ns})}$ for multiples of $n = 6N$ which is $Z_{\mathcal{N}_n}^{(\mathrm{ns})} = \sum_k |\lambda_k|^n$. Substituting (69) for $b$ and $a$ in (25) and (26), we get

$$
P(|\lambda|) = \delta(\lambda_M - |\lambda|) + \sqrt{\frac{3}{2}} \frac{b\theta(\lambda_M - |\lambda|)}{|\lambda|\xi} I_1(\sqrt{6}\xi), \quad (73a)
$$

$$
n(|\lambda|) = I_0(\sqrt{6}\xi), \quad (73b)
$$

where

$$
\xi = \sqrt{b\ln|\lambda_M/\lambda|}, \quad (74)
$$

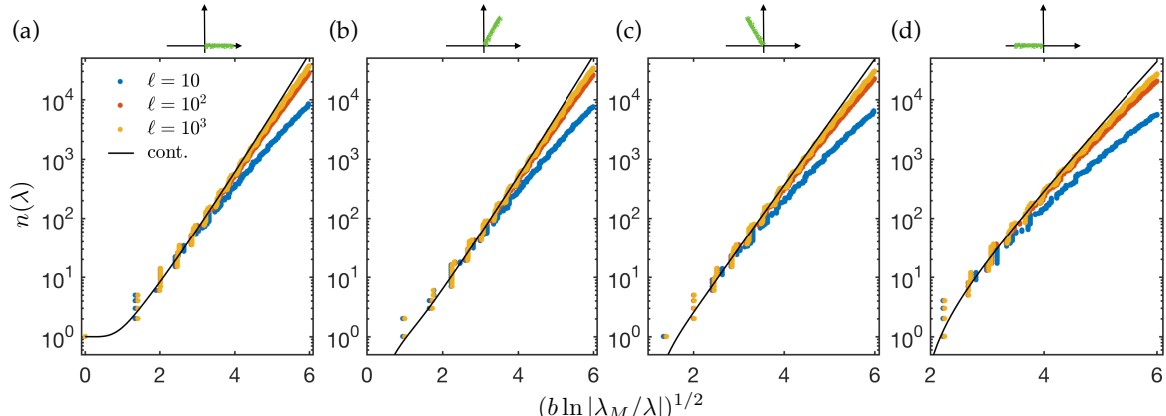

Figure 4: Spectrum of eigenvalues of $\rho^{T_A}$ with a certain complex phase (c.f. Fig. 3(a)) for two equal intervals on an infinite chain. Solid lines are the prediction in Eq. (76b). Dots are numerics, with different colors corresponding to different subsystem sizes. We use the same numerical procedure as in Fig. 3 to obtain few thousand largest (in modulus) many-body eigenvalues from a truncated set of single particle eigenvalues.

and $\lambda_M$ is the largest eigenvalue given by

$$b = -\ln \lambda_M = \lim_{n \to \infty} \frac{1}{n} \ln \mathrm{Tr}(\rho^{T_A})^n = \frac{1}{3} \ln \ell. \tag{75}$$

Figure 3(c) shows a good agreement between the analytical formula (73b) and the numerically obtained spectra for various subsystem sizes. We should note that there is no fitting parameter in (73b) and we only plug in $\lambda_M$ from numerics.

We can further derive the distribution of eigenvalues along different branches in Fig. 3(a). The idea is to analytically continue $Z_{\mathcal{N}_n}^{(\mathrm{ns})}$ with $n = 6N + m$ to arbitrary $n$ and solve the resulting four linearly independent equations generated by (72) to obtain the moments $\sum_j |\lambda_{s_j}|^n$ for each $s = 0, \cdots, 3$. This calculation relies on the assumption that $\lim_{n \to \infty} \frac{r_n}{n}$ does not depend on $m$, which is indeed the case in (69). Hence, we arrive at

$$P_\alpha(\lambda) = \delta(\lambda_M - \lambda)\delta_{\alpha 0} + \frac{b\theta(\lambda_M - |\lambda|)}{6|\lambda|\xi} \sum_{\beta=1}^{2} [M_{\alpha\beta} a_\beta I_1(2a_\beta \xi) - \widetilde{M}_{\alpha\beta} \tilde{a}_\beta J_1(2\tilde{a}_\beta \xi)], \tag{76a}$$

$$n_\alpha(\lambda) = \frac{1}{6} \left[ \sum_{\beta=1}^{2} M_{\alpha\beta} I_0(2a_\beta \xi) + \widetilde{M}_{\alpha\beta} J_0(2\tilde{a}_\beta \xi) \right], \tag{76b}$$

where $P_\alpha(\lambda)$ and $n_\alpha(\lambda)$, $\alpha = 0, \cdots, 3$ describe the distribution of eigenvalues along the $\angle\lambda = \alpha\pi/3$ branch. Here, $M$ and $\widetilde{M}$ encapsulate the coefficients

$$(M|\widetilde{M}) = \begin{pmatrix} 1 & 2 & 2 & 1 \\ 1 & 1 & -1 & -1 \\ 1 & -1 & -1 & 1 \\ 1 & -2 & 2 & -1 \end{pmatrix}, \tag{77}$$

$(a_1, a_2, \tilde{a}_1, \tilde{a}_2) = (\sqrt{\frac{3}{2}}, 1, \frac{1}{\sqrt{2}}, \sqrt{3})$, and $\xi$ and $b$ are defined in Eqs. (74) and (75), respectively. Several comments regarding the phase-resolved distributions (76a) and (76b) are in order. The largest eigenvalue $\lambda_M > 0$ is located on the real axis and hence only appears in $P_0(\lambda)$.

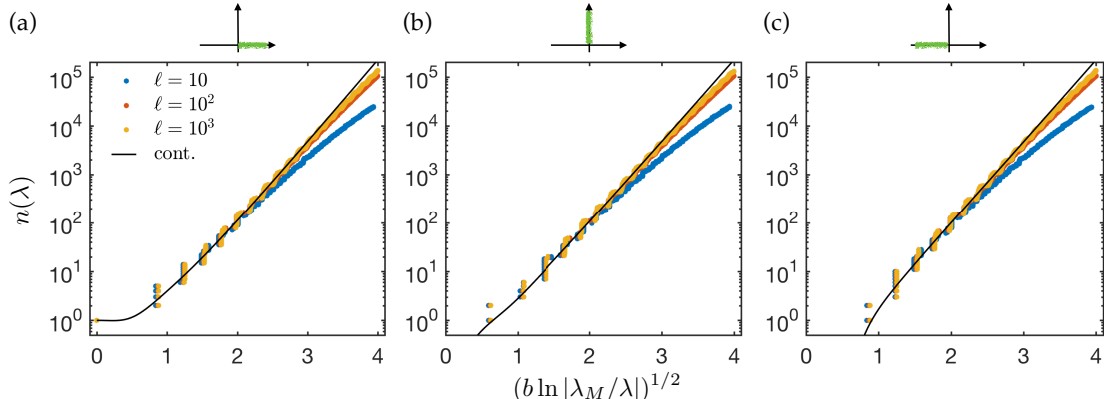

Figure 5: Spectrum of modulus of eigenvalues of $\rho^{T_A}$ for bipartite geometry along the real and imaginary axes. The total system size is $L = 2\ell$ for each $\ell$. Solid lines are the prediction in Eq. (85b). Dots are numerics, with different colors corresponding to different subsystem sizes. A numerical procedure similar to that of Fig. 3 is used to obtain few thousand largest (in modulus) many-body eigenvalues from a truncated set of single particle eigenvalues.

The distribution of modulus is found by $(P_0 + 2P_1 + 2P_2 + P_3)$ which reproduces (73a). It is easy to check that the distribution is normalized and consistent with the identity $\text{Tr}\rho^{T_A} = 1$,

$$\int \lambda P(\lambda) d\lambda = \int \lambda [P_0(\lambda) + P_1(\lambda) - P_2(\lambda) - P_3(\lambda)] d\lambda$$

$$= \int_0^{\lambda_M} \lambda [\delta(\lambda_M - \lambda) + \frac{a_2}{\lambda \xi_2} I_1(2\xi_2)] d\lambda = 1. \tag{78}$$

It is also possible to study the scaling of the maximum eigenvalue (in modulus) $|\lambda_M^{(\alpha)}|$ along each branch. For the bosonic negativity, there are only two branches (positive and negative real axis) and it was found that the scaling of the maxima is the same in the thermodynamic limit [72]. In our case, for a given branch (labeled by $\alpha$) the maximum $|\lambda_M^{(\alpha)}|$ (with $|\lambda_M^0| \equiv \lambda_M$) can be extracted as

$$\ln|\lambda_M^{(\alpha)}| = \lim_{n \to \infty} \frac{1}{n} \ln \sum_j |\lambda_j^{(\alpha)}|^n = -b, \tag{79}$$

where the result is independent of $\alpha$. This again implies the same scaling along each branch, up to a possible unknown constant due to non-universal coefficient that we are dropping in the above formulas (see Eq. (24)).

We compare the analytical results with the numerical simulations for each branch in Fig. 4. As expected, the numerical spectra reach the continuum field theory calculations as we make the system larger. We should point out that in contrast with the bosonic negativity spectrum and the entanglement spectrum which are given solely in terms of $I_\alpha(x)$, the modified Bessel function of the first kind, here the fermionic negativity spectrum contains the Bessel functions $J_\alpha(x)$ as well. Recall that unlike $I_\alpha(x)$ which is strictly positive for $x > 0$, $J_\alpha(x)$ does oscillate between positive and negative values. Nevertheless, there is no issue in $P_\alpha(\lambda)$ which has to be non-negative, as the linear combinations of $I_\alpha$ and $J_\alpha$ in (76b) are such that they are strictly positive over their range of applicability within each branch.

### 4.1.2 Bipartite geometry

Here, we consider two intervals which make up the entire system as shown in the upper panel of Fig. 2(b). In this case, the branch points are identified pairwise as $u_A = v_B$ and $v_A = u_B$, where $\ell_1 = v_A - u_A$. The partition functions in momentum space are found to be

$$Z_{k,n}^{(ns)} = \begin{cases} \ell_1^{-8\frac{k^2}{n^2}} & |k/n| < 1/4, \\ \ell_1^{-2(2|\frac{k}{n}|-1)^2} & |k/n| > 1/4. \end{cases} \tag{80}$$

Similar to the adjacent intervals, the discontinuity in the $k$-dependence comes from the $2\pi$ ambiguity of the $U(1)$ monodromy (Appendix A). As a result, we have $\mathcal{N}_n^{(ns)} = c_n \ln(\ell_1) + \cdots$ where

$$c_n = \begin{cases} -\frac{1}{6}\left(n - \frac{4}{n}\right) & n = 4N, \\ -\frac{1}{6}\left(n - \frac{1}{n}\right) & n = 2N + 1, \\ -\frac{1}{6}\left(n + \frac{8}{n}\right) & n = 4N + 2. \end{cases} \tag{81}$$

A benchmark of these expressions against the scaling of RN in numerical simulations is shown in Fig. 5(c). Because of the cyclic analyticity of the $\mathcal{N}_n^{(ns)}$ modulo four, we expect to have the many-body eigenvalues along the real and imaginary axes. In other words, the complex phase of eigenvalues are multiples of $2\pi/4$.

We now derive the complex phase structure of many-body eigenvalues from the single particle spectrum. In the current case, the density matrix is pure leading to the identity $\gamma^2 = \mathbb{I}$ for the covariance matrix. This property implies that the spectrum of the transformed covariance matrix (40) can be fully determined by the covariance matrix associated to the subsystem A, i.e., $\gamma_{AA}$ in Eq. (40). Hence, the single particle eigenvalues are given by

$$\nu_k = \mu_k + i\sqrt{1 - \mu_k^2}, \tag{82}$$

and its Hermitian conjugate for $\nu_k^*$, where $\mu_k$'s ($k = 0, \cdots, N_A$) denote the eigenvalues of $\gamma_{AA}$ [104]. Using (41), the many-body eigenvalues can be written as

$$\lambda_{\sigma,\sigma'} = \prod_{\sigma_k = \sigma_k'} \frac{1 + \sigma_k \mu_k}{2} \prod_{\sigma_k = -\sigma_k'} \frac{\sigma_k i \sqrt{1 - \mu_k^2}}{2}. \tag{83}$$

This decomposition has two types of factors: real positive and pure imaginary. Therefore, the many-body eigenvalues manifestly lie on the real and imaginary axes. Moreover, the many-body spectrum contains pairs of pure imaginary eigenvalues $\pm i\lambda_j$. The real negative eigenvalues are also two-fold degenerate since they are obtained from the product of even number of pure imaginary factors. In contrast, the real positive eigenvalues are not necessarily degenerate. As a result, the moments of $\rho^{T_A}$ take now the following form

$$Z_{\mathcal{N}_n}^{(ns)} = \sum_j \lambda_{0j}^n + 2\cos\left(\frac{\pi n}{2}\right)\sum_j |\lambda_{1j}|^n + \cos(n\pi)\sum_j |\lambda_{2j}|^n, \tag{84}$$

where $\{\lambda_{\alpha j}\}, \alpha = 0, 1, 2$ denote the eigenvalues along $\angle\lambda = \alpha\pi/2$. This expression in turn implies that there are three types of combinations of different branches for all $n$, which is again consistent with (81). By analytically continuing the three cases, we derive the moment

$\sum_j |\lambda_{\alpha j}|^n$ for each branch. The resulting distributions are found to be

$$P_\alpha(\lambda) = \delta(\lambda_M - \lambda)\delta_{\alpha 0} + \frac{b\theta(\lambda_M - |\lambda|)}{4|\lambda|\xi}\left[\sum_{\beta=1}^{2} M_{\alpha\beta}a_\beta I_1(2a_\beta\xi) - \widetilde{M}_\alpha\tilde{a}J_1(2\tilde{a}\xi)\right], \quad (85a)$$

$$n_\alpha(\lambda) = \frac{1}{4}\left[\sum_{\beta=1}^{2} M_{\alpha\beta}I_0(2a_\beta\xi) + \widetilde{M}_\alpha J_0(2\tilde{a}\xi)\right], \quad (85b)$$

where $M$ and $\widetilde{M}$ encode the coefficients

$$(M|\widetilde{M}) = \begin{pmatrix} 1 & 2 & \big| & 1 \\ 1 & 0 & \big| & -1 \\ 1 & -2 & \big| & 1 \end{pmatrix}, \quad (86)$$

$(a_1, a_2, \tilde{a}) = (2, 1, 2\sqrt{2})$ and $\xi$ is defined in (74) with $b = -\ln\lambda_M = \frac{1}{6}\ln\ell$. As shown in Fig. 5, the above formulas are in decent agreement with numerical results.

Also in this case the maximum (in modulus) $|\lambda_M^\alpha|$ along the different branches can be evaluated through Eq. (79), giving (up to an unknown non-universal constant) $\ln|\lambda_M^\alpha| = -b$ independent of $\alpha$. Finally, also for the bipartite geometry, a consistency check is obtained from $\mathrm{Tr}\rho^{T_A} = 1$, which simply follows from a calculation analogous to Eq. (78).

## 4.2 Spectrum of $\rho^{\widetilde{T}_A}$

Again, the first step to find the moments is the momentum decomposition of (50), yielding

$$Z_{\mathcal{N}_n}^{(\mathrm{r})} = \prod_{k=-(n-1)/2}^{(n-1)/2} Z_{k,n}^{(\mathrm{r})}, \quad (87)$$

where the partition function

$$Z_{k,n}^{(\mathrm{r})} = \langle \widetilde{\mathcal{T}}_{k,n}^{-1}(u_A)\widetilde{\mathcal{T}}_{k,n}(v_A)\mathcal{T}_{k,n}(u_B)\mathcal{T}_{k,n}^{-1}(v_B)\rangle. \quad (88)$$

is subject to modified monodromy conditions for the $\widetilde{\mathcal{T}}_{k,n}$ and $\widetilde{\mathcal{T}}_{k,n}^{-1}$, which are $\psi_k \mapsto e^{\pm i(2\pi k/n - \pi)}\psi_k$. This monodromy is different from the supersymmetric trace [118] (see Appendix C for the definition and more details).

### 4.2.1 Adjacent intervals

In this case, we find that

$$Z_{k,n}^{(\mathrm{r})} \propto \ell_1^{-2(|\frac{k}{n}|-\frac{1}{2})(|\frac{2k}{n}|-\frac{1}{2})} \cdot \ell_2^{-2|\frac{k}{n}|(|\frac{2k}{n}|-\frac{1}{2})} \cdot (\ell_1+\ell_2)^{2|\frac{k}{n}|(|\frac{k}{n}|-\frac{1}{2})}. \quad (89)$$

It is important to note that for $k < 0$, we modified the flux at $u_1$ and $v_1$ by inserting additional $2\pi$ and $-2\pi$ fluxes, respectively, where the scaling exponent takes its minimum value (c.f. Appendix A). Summing up $Z_{k,n}^{(\mathrm{r})}$ terms, we get

$$\mathcal{N}_n^{(\mathrm{r})} = c_n^{(1)}\ln(\ell_1) + c_n^{(2)}\ln(\ell_2) + c_n^{(3)}\ln(\ell_1+\ell_2) + \cdots, \quad (90)$$

where

$$c_{n_o}^{(1)} = -\frac{1}{12}\left(n_o + \frac{5}{n_o}\right), \quad (91)$$

$$c_{n_o}^{(2)} = c_{n_o}^{(3)} = -\frac{1}{12}\left(n_o - \frac{1}{n_o}\right), \quad (92)$$

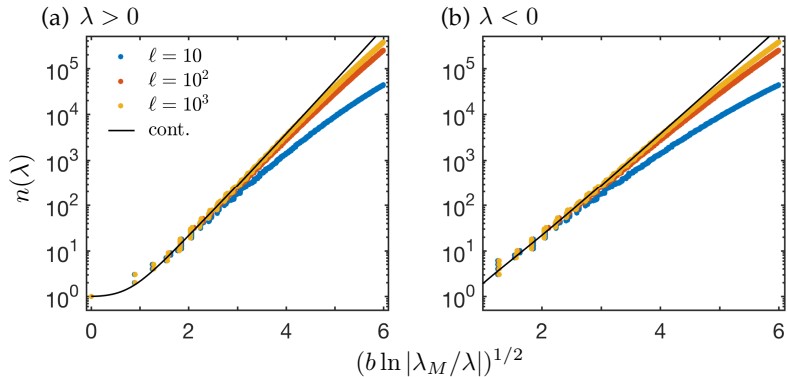

Figure 6: Tail distribution function for the spectrum of $\rho^{\widetilde{T}_A}$ of two equal adjacent intervals on an infinite chain. Solid lines are the analytical distributions from Eq. (97b). Dots are numerics, with different colors corresponding to different subsystem sizes. We use the same numerical procedure as in Fig. 3 to obtain few thousand largest (in modulus) many-body eigenvalues from a truncated set of single particle eigenvalues.

for odd $n = n_o$, and

$$c_{n_e}^{(1)} = c_{n_e}^{(2)} = -\frac{1}{6}\left(\frac{n_e}{2} - \frac{2}{n_e}\right), \tag{93}$$

$$c_{n_e}^{(3)} = -\frac{1}{6}\left(\frac{n_e}{2} + \frac{1}{n_e}\right), \tag{94}$$

for even $n = n_e$. As a consistency check, we show in Appendix B that the above formulae can be derived from two disjoints intervals as the distance between the intervals is taken to be zero. Notice that the even $n$ case is identical to the general CFT results [70]. Also, from (19) we arrive at the familiar result for the logarithmic negativity,

$$\mathcal{E} = \frac{1}{4}\ln\left(\frac{\ell_1\ell_2}{\ell_1 + \ell_2}\right) + \cdots. \tag{95}$$

For equal length intervals, we may write $\mathcal{N}_n^{(\mathrm{r})} = c_n \ln\ell + \cdots$ where

$$c_n = \begin{cases} -\frac{1}{4}\left(n_o + \frac{1}{n_o}\right) & n = n_o \quad \text{odd,} \\ -\frac{1}{2}\left(\frac{n_e}{2} - \frac{1}{n_e}\right) & n = n_e \quad \text{even.} \end{cases} \tag{96}$$

As expected for Hermitian operator $\rho^{\widetilde{T}_A}$, here the moments $\mathcal{N}_n^{(\mathrm{r})}$ only depend on parity of $n$, i.e., whether $n$ is odd or even. This means that the eigenvalues are real positive or negative. We can also see this from the fact that the single particle spectrum is real. The many-body eigenvalues follow the form $\lambda_{\boldsymbol{\sigma}} = \prod_{\sigma_k = \pm}(1 + \sigma_k \nu_k)/2$, where $\nu_k$ are single-particle eigenvalues of the covariance matrix (42). As discussed in the previous section, we carry out the same procedure to derive the distribution from analytic continuation of moments (in this case there are only two branches). The final result reads

$$P(\lambda) = \delta(\lambda_M - \lambda) + \frac{b\theta(\lambda_M - |\lambda|)}{2|\lambda|\xi}[-J_1(2\xi)\mathrm{sgn}(\lambda) + \sqrt{2}I_1(2\sqrt{2}\xi)], \tag{97a}$$

$$n(\lambda) = \frac{1}{2}[J_0(2\xi)\mathrm{sgn}(\lambda) + I_0(2\sqrt{2}\xi)], \tag{97b}$$

where $\xi$ obeys the same form as Eq. (74) with a slight difference that $b = -\ln\lambda_M = \frac{1}{4}\ln\ell$. We present a comparison of the above expression with numerical spectrum of free fermions

on the lattice of different lengths in Fig 6. There is a good agreement between analytical and numerical results.

We further find that, as it was the case for the bosonic negativity, the scaling of the minimum and maximum eigenvalue is the same. Finally, we confirm that the distribution probability is properly normalized such that $\int \lambda P(\lambda) d\lambda = \text{Tr}[\rho(-1)^{F_A}]$ and it is consistent with $\mathcal{E} = \frac{1}{4}\ln\ell$, Eq. (19), which follows from

$$\mathcal{E} = \ln \int d\lambda \, |\lambda| P(\lambda) = \ln\left[\lambda_M + \int_0^{\lambda_M} d\lambda \, \frac{b\sqrt{2}}{\xi} I_1(2\sqrt{2}\xi)\right] = \frac{1}{4}\ln\ell. \tag{98}$$

### 4.2.2 Bipartite geometry

In this case, we start by computing the correlator

$$Z_{k,n}^{(\text{r})} = \langle \mathcal{Q}_{k,n}^{-2}(u_A) \mathcal{Q}_{k,n}^{2}(v_A) \rangle \propto \ell_1^{-2(|\frac{2k}{n}|-\frac{1}{2})^2}. \tag{99}$$

Here again, we have to minimize the scaling exponent for $k < 0$ by inserting additional $2\pi$ fluxes (c.f. Appendix A). The RN is then found to be $\mathcal{N}_n^{(\text{r})} = c_n \ln(\ell_1) + \cdots$ where

$$c_n = \begin{cases} -\frac{1}{6}\left(n_o + \frac{2}{n_o}\right) & n = n_o \quad \text{odd}, \\ -\frac{1}{3}\left(\frac{n_e}{2} - \frac{2}{n_e}\right) & n = n_e \quad \text{even}. \end{cases} \tag{100}$$

From this, we derive the distribution of many-body eigenvalues to be

$$P(\lambda) = \delta(\lambda_M - \lambda) + \frac{b\theta(\lambda_M - |\lambda|)}{2|\lambda|\xi}[-\sqrt{2}J_1(2\sqrt{2}\xi)\text{sgn}(\lambda) + 2I_1(4\xi)], \tag{101a}$$

$$n(\lambda) = \frac{1}{2}[J_0(2\sqrt{2}\xi)\text{sgn}(\lambda) + I_0(4\xi)], \tag{101b}$$

where $\xi$ is given in (74) and $b = -\ln\lambda_M = \frac{1}{6}\ln\ell$.

We finish this part by a remark about the covariance matrix. Using the fact that $\gamma^2 = \mathbb{I}$ for pure states, the covariance matrix (42) can be further simplified into

$$\widetilde{\gamma} = \begin{pmatrix} \gamma_{AA} - 2\gamma_{AA}^{-1} & -i\gamma_{AB} \\ i\gamma_{BA} & \gamma_{BB} \end{pmatrix}. \tag{102}$$

Similar to the adjacent intervals, we can calculate the many-body spectrum out of eigenvalues of the above covariance matrix. We confirm that the numerical results and analytical expressions match. However, we avoid showing the plots here as they look quite similar to Fig. 6.

## 5 Conclusions

In summary, we study the distribution of the eigenvalues of partially transposed density matrices, aka the negativity spectrum, in free fermion chains. Taking the PT of fermionic density matrices is known to be a difficult task even for free fermions (or Gaussian states). However, recent studies [100, 101, 112] suggest that this difficulty could be circumvented if we use a different definition for partial transpose which is closely related to time-reversal transformation. In a matrix representation of a fermionic density matrix, e.g. in Fock space basis, the latter operation involves multiplying a $\mathbb{Z}_4$ complex phase factor in addition to the matrix transposition where the phase factor solely depends on the fermion-number parity of the state of

subsystems to be exchanged in the transpose process. It turned out that the phase factor in the fermionic partial transpose lead to two types of partial transpose operation $\rho^{T_A}$ and $\rho^{\widetilde{T}_A}$. The difference is that $\rho^{T_A}$ is pseudo-Hermitian and may contain complex eigenvalues, while $\rho^{\widetilde{T}_A}$ is Hermitian and its eigenvalues are real. This is in contrast with the fact that the standard partial transpose $\rho^{T_A}$ is always a Hermitian operator which implies a real spectrum. In this paper, we presented analytical and numerical results for the negativity spectra of two adjacent intervals on a free fermion chain using both types of fermionic partial transpose. In the case of $\rho^{\widetilde{T}_A}$, we find that the negativity spectra share a lot of similarities with those found in a previous CFT work [72]. However, in the case of $\rho^{T_A}$, we realize that the eigenvalues form a special pattern on the complex plane and fall on six branches with a quantized phase of $2\pi n/6$. The spectrum in the latter case is mirror symmetric with respect to the real axis, and there are four universal functions which describe the distributions along the six branches. The sixfold distribution of eigenvalues is not specific to complex fermion chain (described by the Dirac Hamiltonian) with $c = 1$ and also appears in the critical Majorana chain with $c = 1/2$. We further confirmed that our analytical expressions are applicable to the Majorana chain upon modifying the central charge $c$. The fact that the negativity of two adjacent intervals is given by three-point correlators of twist fields suggests that our free fermion results for the negativity spectrum may be generalized to other (possibly interacting) fermionic CFTs. Our method is also applicable to the case of two disjoint intervals; however, the result may not be as universal since the negativity of two disjoint intervals involves four-point correlator of twist fields (which depends on the full operator content of CFTs).

Given our free fermion results in one dimension, there are several avenues to pursue for future research. A natural extension is to explore possible structures in the negativity spectrum of free fermions in higher dimensions. It would also be interesting to understand the effect of disorder and spin-orbit coupling on this distribution. In particular, the random singlet phase (RSP) [119], which can be realized in the strongly disordered regime of one-dimensional free fermions, is characterized by logarithmic entanglement entropy [63, 120, 121] that is a hallmark of $(1 + 1)$d critical theories. An interesting question is how the negativity spectrum of critical RSP differs from the clean limit which was studied in this paper. Another direction could be studying strongly correlated fermion systems and specially interacting systems which have a description in terms of projected free fermions such as the Haldane-Shastry spin chain [122, 123]. Furthermore, it is worth investigating how thermal fluctuations affect the negativity spectrum in finite-temperature states. Finally, the negativity spectrum may be useful in studying the quench dynamics and shed light on thermalization.

## Acknowledgments

The authors would like to acknowledge insightful discussions with Erik Tonni, David Huse, Zoltan Zimboras.

**Funding information** SR and HS were supported in part by the National Science Foundation under Grant No. DMR-1455296, and under Grant No. NSF PHY-1748958. PC and PR acknowledge support from ERC under Consolidator grant number 771536 (NEMO). We all thank the Galileo Galilei Institute for Theoretical Physics for the hospitality and the INFN for partial support during the completion of this work. H.S. acknowledges the support from the ACRI fellowship (Italy) and the KITP graduate fellowship program. SR is supported by a Simons Investigator Grant from the Simons Foundation.

# A  Twist fields, bosonization, etc.

The Rényi entanglement entropy (REE) of a reduced density matrix $\rho$ is defined in Eq. (2). For non-interacting systems with conserved $U(1)$ charge, we can transform the trace formulas into a product of $n$ decoupled partition functions. Let us first illustrate this idea for REE [114]. We can diagonalize the twist matrix $T$ in Eq. (45) and rewrite the REE in terms of $n$-decoupled copies,

$$Z_{\mathcal{R}_n} = \int \prod_k d\psi_k d\bar{\psi}_k \prod_k \left[\rho(\bar{\psi}_k, \psi_k)\right] e^{\sum_k \lambda_k \bar{\psi}_k \psi_k}, \tag{103}$$

where $\lambda_k = e^{i2\pi\frac{k}{n}}$ for $k = (n-1)/2, \cdots, (n-1)/2$ are eigenvalues of the twist matrix. In this new basis, the transformation rule $\Psi \to T\Psi$ for the field passing through the interval becomes a phase twist, i.e., $\psi_k \mapsto \lambda_k \psi_k$. Therefore, the REE can be decomposed into product of separate factors as

$$Z_{\mathcal{R}_n} = \prod_{k=-(n-1)/2}^{(n-1)/2} Z_{k,n}, \tag{104}$$

where $Z_{k,n}$ is the partition function containing an interval with the twisting phase $2\pi k/n$. We reformulate the partition function in the presence of phase twisting intervals in terms of a theory subject to an external gauge field which is a pure gauge everywhere (except at the points $u_i$ and $v_i$ where it is vortex-like). This is obtained by a singular gauge transformation

$$\psi_k(x) \to e^{i\int_{x_0}^{x} dx'^{\mu} A_{\mu}^{k}(x')} \psi_k(x), \tag{105}$$

where $x_0$ is an arbitrary fixed point. Hence, for a subsystem made of $p$ intervals, $A = \bigcup_{i=1}^{p} [u_i, v_i]$, we can absorb the boundary conditions across the intervals into an external gauge field and the resulting Lagrangian density becomes

$$\mathcal{L}_k = \bar{\psi}_k \gamma^{\mu} \left(\partial_{\mu} + i A_{\mu}^{k}\right) \psi_k, \tag{106}$$

where the $U(1)$ flux is given by

$$\epsilon^{\mu\nu}\partial_{\nu}A_{\mu}^{k}(x) = 2\pi\frac{k}{n}\sum_{i=1}^{p}\left[\delta(x-u_i) - \delta(x-v_i)\right]. \tag{107}$$

Note that there is an ambiguity in the flux strength, namely, $2\pi m$ (integer $m$) fluxes may be added to the right hand side of the above expression, while the monodromy for the fermion fields does not change. To preserve this symmetry (or redundancy), $Z_k$ must be written as a sum over all representations [124–127]. The asymptotic behavior of each term in this expansion is a power law $\ell^{-\alpha_m}$ in thermodynamic limit (large (sub-)system size). Here, we are interested in the leading order term which corresponds to the smallest exponent $\alpha_m$.

As we will see in the case of entanglement negativity, we need to consider $m \neq 0$ for some values of $k$. Let us first discuss this expansion for a generic case. Let $\mathcal{S}_n$ be a partition function on a multi-sheet geometry (for either Rényi entropy or negativity). As mentioned, after diagonalizing the twist matrices $\mathcal{S}_n$ can be decomposed as

$$\mathcal{S}_n = \sum_k \ln Z_k, \tag{108}$$

where $Z_k$ is the partition function in the presence of $2p$ flux vortices at the two ends of $p$ intervals between $u_{2i-1}$ and $u_{2i}$, that is

$$Z_k = \left\langle e^{i \int A_{k,\mu} j_k^\mu d^2 x} \right\rangle, \tag{109}$$

in which

$$\epsilon^{\mu\nu} \partial_\nu A_{k,\mu}(x) = 2\pi \sum_{i=1}^{2p} \nu_{k,i} \delta(x - u_i), \tag{110}$$

and $2\pi \nu_{k,i}$ is vorticity of gauge flux determined by the eigenvalues of the twist matrix. The total vorticity satisfies the neutrality condition $\sum_i \nu_{k,i} = 0$ for a given $k$. In order to obtain the asymptotic behavior, one needs to take the sum over all the representations of $Z_k$ (i.e., flux vorticities mod $2\pi$),

$$Z_k = \sum_{\{m_i\}} Z_k^{(m)}, \tag{111}$$

where $\{m_i\}$ is a set of integers and

$$Z_k^{(m)} = \left\langle e^{i \int A_{k,\mu}^{(m)} j_k^\mu d^2 x} \right\rangle, \tag{112}$$

is the partition function for the following fluxes,

$$\epsilon^{\mu\nu} \partial_\nu A_\mu^{(m),k}(x) = 2\pi \sum_{i=1}^{2p} \widetilde{\nu}_{k,i} \delta(x - u_i), \tag{113}$$

and $\widetilde{\nu}_{k,i} = \nu_{k,i} + m_i$ are shifted flux vorticities. The neutrality condition requires $\sum_i m_i = 0$. Using the bosonization technique, we obtain

$$Z_k^{(m)} = C_{\{m_i\}} \prod_{i<j} |u_i - u_j|^{2\widetilde{\nu}_{k,i}\widetilde{\nu}_{k,j}}, \tag{114}$$

where $C_{\{m_i\}}$ is a constant depending on cutoff and microscopic details. We make use of the neutrality condition $-2\sum_{i<j} \widetilde{\nu}_{k,i}\widetilde{\nu}_{k,j} = \sum_i \widetilde{\nu}_{k,i}^2$ and rewrite

$$Z_k \sim \sum_{\{m_i\}} C_{\{m_i\}} \ell^{2\sum_{i<j} \widetilde{\nu}_{k,i}\widetilde{\nu}_{k,j}} = \sum_{\{m_i\}} C_{\{m_i\}} \ell^{-\sum_i \widetilde{\nu}_{k,i}^2}, \tag{115}$$

where $\ell$ is a length scale. From this expansion, the leading order term in the limit $\ell \to \infty$ is clearly the one(s) which minimizes the quantity $\sum_i \widetilde{\nu}_{k,i}^2$. This is identical to the condition derived from the generalized Fisher-Hartwig conjecture [128, 129]. A careful determination of the leading order term for REE by a similar approach was previously discussed in Ref. [125–127].

We now carry out this process for $Z_{\mathcal{N}_n}^{(\text{ns})}$ in Eq. (67) for two adjacent intervals. Here, we need to minimize the quantity

$$f_{m_1 m_2 m_3}(\nu) = (\nu + m_1)^2 + (\nu + m_3)^2 + (-2\nu + m_2)^2 \tag{116}$$

for a given $\nu = k/n = -(n-1)/2n, \cdots, (n-1)/2n$ by finding the integers $(m_1, m_2, m_3)$ constrained by $\sum_i m_i = 0$. For instance, let us compare $(0,0,0)$ with $(-1,1,0)$,

$$f_{000}(\nu) = 6\nu^2, \tag{117}$$

$$f_{-110}(\nu) = 6\nu^2 - 6\nu + 2. \tag{118}$$

So, we have

$$f_{000}(\nu) > f_{-110}(\nu) \qquad \text{for} \quad \nu > \frac{1}{3}. \tag{119}$$

Similarly, we find that

$$f_{000}(\nu) > f_{1-10}(\nu) \qquad \text{for} \quad \nu < -\frac{1}{3}. \tag{120}$$

In summary, we resolve the flux ambiguity by adding the triplet $(m_1, m_2, m_3)$ as follows

$$\begin{cases} (0,0,0) & |\nu| \le 1/3 \\ (-1,1,0),(0,1,-1) & \nu > 1/3 \\ (1,-1,0),(0,-1,1) & \nu < -1/3. \end{cases} \tag{121}$$

This leads us to write Eq. (68). Finally, similar derivation can be carried out to arrive at Eqs. (80), (89) and (99).

# B    Rényi negativity for disjoint intervals

In this appendix, we derive the RN associated with $\rho^{T_A}$ and $\rho^{\widetilde{T_A}}$ for two disjoint intervals and show that upon taking the distance between the intervals to zero, we recover the results for two adjacent intervals as discussed in the main text.

## B.1    Moments of $\rho^{T_A}$

This geometry is characterized by $v_A - u_A = \ell_1$, $u_B - v_A = d$, and $v_B - u_B = \ell_2$ (c.f. Fig. 7(a)). The leading order term of the momentum decomposed partition function in the case of disjoint intervals is given by

$$Z_{k,n}^{(\text{ns})} = c_{k0} \left( \frac{x}{\ell_1 \ell_2} \right)^{2k^2/n^2} + \cdots, \tag{122}$$

where

$$x = \frac{(\ell_1 + \ell_2 + d)d}{(\ell_1 + d)(\ell_2 + d)}. \tag{123}$$

Consequently, the RN is found to be

$$\mathcal{N}_n^{(\text{ns})} = \left( \frac{n^2 - 1}{6n} \right) \ln \left( \frac{x}{\ell_1 \ell_2} \right) + \cdots. \tag{124}$$

We compare the above formula with the scaling behavior of the numerical results in Fig. 7(b), where we find that they match.

As a consistency check, we show that the RN between adjacent intervals can be derived as a limiting behavior of the disjoint intervals. However, we realize from (122) that $\lim_{d \to 0} Z_{k,n}^{(\text{ns})} = 0$ (as is done also in Ref. [110]). A more careful treatment goes by considering higher order terms coming from different representations in (115)

$$Z_{k,n}^{(\text{ns})} = c_{k0} \left( \frac{x}{\ell_1 \ell_2} \right)^{2\frac{k^2}{n^2}} + c_{k1} \left( \frac{x}{\ell_1 \ell_2} \right)^{2|\frac{k}{n}|(|\frac{k}{n}|-1)} [g(\ell_1, \ell_2; k/n) + g(\ell_1 + d, \ell_2 + d; k/n)] + \cdots, \tag{125}$$

where $g(x, y; q) = x^{-2}(x/y)^{2|q|} + x \leftrightarrow y$ and $c_{ki}$ are coefficients dependent on the microscopic details. Next, we obtain the leading order term in the coincident limit $d = \varepsilon$, where $\varepsilon \ll \ell_1, \ell_2$. To this end, we rewrite the above expansion (125) as

$$Z_{k,n}^{(\text{ns})} = \varepsilon^{2k^2/n^2} Z_{k,n}^{(0)} + \varepsilon^{2|k/n|(|k/n|-1)} Z_{k,n}^{(1)} + \cdots, \tag{126}$$

where the scaling dimensions are

$$[Z_{k,n}^{(0)}] \sim L^{-6N^2/n^2}, \tag{127a}$$

$$[Z_{k,n}^{(1)}] \sim L^{-2(3k^2/n^2 - 3|k/n| + 1)}. \tag{127b}$$

As we see, for $|k/n| > 1/3$, the second term is dominant. This immediately implies that upon taking $(\ell_i + d) \sim \ell_i$, we recover the original result (68).

## B.2 Moments of $\rho^{\widetilde{T}_A}$

Similarly, we find the $k$-th contribution to the $n$-th moment of $\rho^{\widetilde{T}_A}$ to be

$$Z_{k,n}^{(\text{r})} = x^{2|k/n|(|k/n|-1/2)} \frac{1}{\ell_1^{2(|k/n|-1/2)^2} \ell_2^{2k^2/n^2}} + \cdots, \tag{128}$$

which gives rise to the following form for the RN,

$$\mathcal{N}_n^{(\text{r})} = c_n^{(1)} \ln(\ell_1) + c_n^{(2)} \ln(\ell_2) + c_n^{(3)} \ln(x) + \cdots, \tag{129}$$

where

$$c_n^{(1)} = \begin{cases} -\frac{1}{6}\left(n_o + \frac{2}{n_o}\right) & n = n_o \quad \text{odd}, \\ -\frac{1}{6}\left(n_e - \frac{1}{n_e}\right) & n = n_e \quad \text{even}, \end{cases} \tag{130}$$

$$c_n^{(2)} = -\left(\frac{n^2 - 1}{6n}\right), \tag{131}$$

$$c_n^{(3)} = \begin{cases} -\frac{1}{12}\left(n_o - \frac{1}{n_o}\right) & n = n_o \quad \text{odd}, \\ -\frac{1}{6}\left(\frac{n_e}{2} + \frac{1}{n_e}\right) & n = n_e \quad \text{even}. \end{cases} \tag{132}$$

We compare the scaling behaviors of analytical expressions and numerical results in Fig. 7(c). As we see, they are in good agreement.

It is easy to verify that taking the adjacent limit $d = \varepsilon$ of two disjoint intervals in Eq. (129) leads to Eq. (90). We should note that in this case the leading order term in the momentum expansion (128) remains always the same (128) in contrast with the previous case (125).

## C Partial transpose with supersymmetric trace

Let

$$\mathcal{N}_n^{(\text{susy})}(\rho) = \ln \widetilde{\text{Tr}}[(\rho^{\widetilde{T}_A})^n], \tag{133}$$

where $\widetilde{\text{Tr}}$ is the supersymmetric (susy) trace for the interval $A$. The susy trace is distinct from the regular trace in that the $T$ matrix which glues together $\rho^{\widetilde{T}_A}$ for fermions is given by (45), while the susy trace is similar to a bosonic trace (even though applied to a fermionic density matrix) where the $T$ matrix is given by (145) (see below). It is easy to see that $T^n = (-1)^{n-1}$

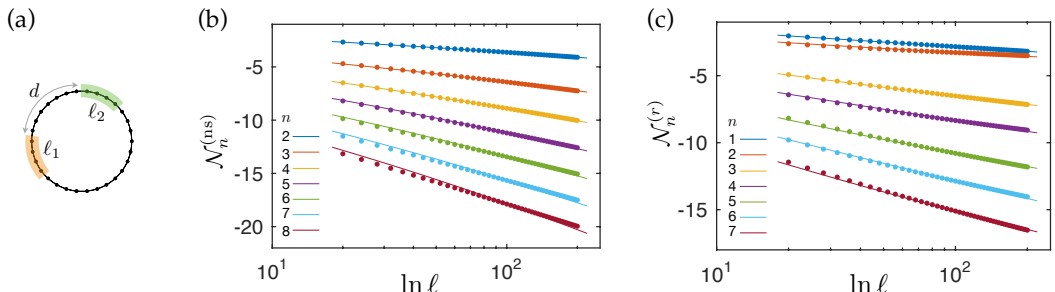

Figure 7: Comparison of numerical (dots) and analytical (solid lines) results for the scaling behavior of the moments of partial transpose (48) and (50) for two disjoint intervals (the geometry is shown in panel (a)). Here, $d = 40$ and intervals have equal lengths $\ell_1 = \ell_2 = \ell$ where $20 \leq \ell \leq 200$ on an infinite chain. The analytical results are given by Eq. (122) in panel (b) and Eq. (128) in panel(c). Different colors correspond to different moments $n$.

for the regular trace of fermionic density matrices whereas $T^n = 1$ for the susy trace. Clearly, there is no difference between the susy and regular traces for even $n$ when considering $(\rho^{\tilde{T}_A})^n$. The susy trace was used previously to define the susy entanglement entropies [118]. (See Refs. [130–132] for related works.) In terms of the partial transpose (13), the susy trace is simplified into

$$\mathcal{N}_n^{(\text{susy})}(\rho) = \begin{cases} \ln \text{Tr}(\rho^{T_A} \rho^{T_A \dagger} \cdots \rho^{T_A} \rho^{T_A \dagger}) & n \text{ even}, \\ \ln \text{Tr}(\rho^{T_A} \rho^{T_A \dagger} \cdots \rho^{T_A}) & n \text{ odd}, \end{cases} \tag{134}$$

which was studied by some of us in [112] and was shown to obey the same expressions as the bosonic negativity [70] for both even and odd values of $n$. In this appendix, we briefly report the results for various geometries. A technical point is that the monodromy of the field around $\widetilde{\mathcal{T}}_{k,n}$ for the susy trace is given by $\psi_k \mapsto e^{\pm i(2\pi k/n - \varphi_n)}\psi_k$ where $\varphi_n = \pi$ or $\pi(n-1)/n$ for $n$ even or odd, respectively [110, 133].

1. **Disjoint intervals:** in this case the moments (133) become

$$\mathcal{N}_n^{(\text{susy})} = c_n^{(1)} \ln(\ell_1 \ell_2) + c_n^{(2)} \ln(x) + \cdots, \tag{135}$$

where $x$ is defined in (123), $c_n^{(1)} = -(n^2 - 1)/6n$, and

$$c_n^{(2)} = \begin{cases} -\frac{1}{12}\left(n_o - \frac{1}{n_o}\right) & n = n_o \quad \text{odd}, \\ -\frac{1}{6}\left(\frac{n_e}{2} + \frac{1}{n_e}\right) & n = n_e \quad \text{even}. \end{cases} \tag{136}$$

2. **Adjacent intervals:** when the distance $d \to 0$, i.e., $x \to 0$ in the above expression, the moments take the form

$$\mathcal{N}_n^{(\text{susy})} = c_n^{(1)} \ln(\ell_1 \ell_2) + c_n^{(2)} \ln(\ell_1 + \ell_2) + \cdots, \tag{137}$$

where

$$c_{n_o}^{(1)} = c_{n_o}^{(2)} = -\frac{1}{12}\left(n_o - \frac{1}{n_o}\right), \tag{138}$$

for odd $n = n_o$, and

$$c_{n_e}^{(1)} = -\frac{1}{6}\left(\frac{n_e}{2} - \frac{2}{n_e}\right), \tag{139}$$

$$c_{n_e}^{(2)} = -\frac{1}{6}\left(\frac{n_e}{2} + \frac{1}{n_e}\right), \tag{140}$$

for even $n = n_e$.

**3. Bipartite geometry:** finally, in this case one has

$$\mathcal{N}_n^{(\mathrm{r})} = c_n \ln(\ell_1) + \cdots, \tag{141}$$

where

$$c_n = \begin{cases} -\frac{1}{6}\left(n_o - \frac{1}{n_o}\right) & n = n_o \quad \text{odd}, \\ -\frac{1}{3}\left(\frac{n_e}{2} - \frac{2}{n_e}\right) & n = n_e \quad \text{even}. \end{cases} \tag{142}$$

## D   Negativity of bosonic scalar field theory

As we have seen in the main text, calculating negativity boils down to computing correlators of twist fields. In this appendix, we briefly review the conformal weights of the twist fields in the complex scalar field theory,

$$\mathcal{L}_\phi = \frac{1}{4\pi}\int |\nabla\phi|^2, \tag{143}$$

from which we can compute the correlators of twist fields and derive expressions for the entanglement of free bosons. Similar to fermions, the moments of density matrix in the coherent basis read as

$$Z_{\mathcal{R}_n} = \mathrm{Tr}[\rho^n] = \int \prod_{i=1}^n d\phi_i d\phi_i^* \prod_{i=1}^n \left[\rho(\phi_i^*, \phi_i)\right] e^{\sum_{i,j}\phi_i^* T_{ij}\phi_j}, \tag{144}$$

where

$$T = \begin{pmatrix} 0 & 1 & 0 & \cdots \\ 0 & 0 & 1 & 0 \\ \vdots & \vdots & \ddots & 1 \\ 1 & 0 & \cdots & 0 \end{pmatrix}. \tag{145}$$

For the moments of the partial transpose, we have

$$Z_{\mathcal{N}_n} = \mathrm{Tr}[(\rho^{T_A})^n] = \int \prod_{i=1}^n d\phi_i d\phi_i^* \prod_{i=1}^n \left[\rho(\phi_i^*, \phi_i)\right] e^{\sum_{i,j}\phi_i^{A*}[T^{-1}]_{ij}\phi_j^A} e^{\sum_{i,j}\phi_i^{B*}T_{ij}\phi_j^B}. \tag{146}$$

In the case of free bosons, the moments can be written as a product of partition functions of decoupled modes,

$$Z_{\mathcal{R}_n} = \prod_{k=0}^{n-1} \langle \mathcal{T}_{k,n}^{-1}(0)\mathcal{T}_{k,n}(\ell)\rangle, \tag{147}$$

$$Z_{\mathcal{N}_n} = \prod_{k=0}^{n-1} \langle \mathcal{T}_{k,n}^{-1}(-\ell_1)\mathcal{T}_{k,n}^2(0)\mathcal{T}_{k,n}^{-1}(\ell_2)\rangle. \tag{148}$$

Hence, our objective here is to find the conformal weight of $\mathcal{T}_{k,n}$, $\mathcal{T}_{k,n}^2$, and their adjoints. It is worth noting that in the case of bosons $k$ takes positive integer values, $k = 0, 1, \cdots, n-1$. This is because the global boundary condition is periodic, i.e. the twist matrix obeys $T^n = 1$. As usual in the conformal field theory, the computation goes by placing a twist field $\mathcal{T}_{k,n}$ at the origin which leads to a ground state $\mathcal{T}_{k,n}(0)|0\rangle$ where $\phi(z)$ and $\phi^*(z)$ are multivalued fields with the boundary conditions $\phi(e^{i2\pi}z) = e^{i2\pi k/n}\phi(z)$ and $\phi^*(e^{i2\pi}z) = e^{-i2\pi k/n}\phi^*(z)$. Next, we compute the correlator

$$\langle \partial_z \phi \, \partial_w \phi^* \rangle_{k/n} := \langle \mathcal{T}_{k,n}^{-1}(\infty)|\partial_z \phi \, \partial_w \phi^*|\mathcal{T}_{k,n}(0)\rangle, \tag{149}$$

to find the expectation value of the energy-momentum tensor via

$$\langle T(z) \rangle_{k/n} = -\lim_{z\to w}\left\langle \frac{1}{2}\partial_z \phi \, \partial_w \phi^* + \frac{1}{(z-w)^2} \right\rangle_{k/n}. \tag{150}$$

Using the fact that

$$T(z)\mathcal{T}_{k,n}(0)|0\rangle \sim \frac{\Delta_{\mathcal{T}_{k,n}}}{z^2}\mathcal{T}_{k,n}(0)|0\rangle + \cdots, \tag{151}$$

we can read off the conformal weight $\Delta_{\mathcal{T}_{k,n}}$.

Let us start with $\mathcal{T}_{k,n}$ and $\mathcal{T}_{k,n}^{-1}$. The correlation function (149) can be directly computed by the mode expansion of $\phi$ field or can be simply derived from the asymptotic behavior $z \to w$ and $z \to 0$ or $w \to \infty$. The result is found to be [134, 135],

$$-\frac{1}{2}\langle \partial_z \phi \, \partial_w \phi^* \rangle_{k/n} = z^{k/n-1}w^{-k/n}\left[\frac{z(1-k/n)+wk/n}{(z-w)^2}\right], \tag{152}$$

which leads to

$$\Delta_{\mathcal{T}_{k,n}} = \Delta_{\mathcal{T}_{k,n}^{-1}} = \frac{k}{2n}\left(1 - \frac{k}{n}\right). \tag{153}$$

We should note that doing this calculation for complex Dirac fermions, instead, leads to $\Delta_{\mathcal{T}_{k,n}} = k^2/2n^2$. So, the Rényi entropies are given by

$$\mathcal{R}_n = \frac{2}{1-n}\sum_{k=0}^{n-1}\Delta_{\mathcal{T}_{k,n}} \cdot \ln\ell = \left(\frac{n+1}{6n}\right)\ln\ell. \tag{154}$$

One can do a similar calculation for $\mathcal{T}_{k,n}^2$. In this case, the boundary condition is $\phi(e^{i2\pi}z) = e^{i4\pi k/n}\phi(z)$. For $k/n < 1/2$, the result is identical to (153) up to replacing $k/n$ by $2k/n$. For $1/2 < k/n < 1$ however, the effective phase shift is $2\pi(2k/n - 1)$ and we need to substitute $k/n$ in (153) by $2k/n - 1$. This result can also be understood from the mode expansion. Consequently, we arrive at

$$\Delta_{\mathcal{T}_{k,n}^2} = \Delta_{\mathcal{T}_{k,n}^{-2}} = \begin{cases} \frac{k}{n}\left(1 - \frac{2k}{n}\right) & \frac{k}{n} \leq \frac{1}{2}, \\ \left(\frac{2k}{n} - 1\right)\left(1 - \frac{k}{n}\right) & \frac{1}{2} \leq \frac{k}{n} < 1. \end{cases} \tag{155}$$

Using the following expression for the moments of partial transpose,

$$\mathcal{N}_n = c_n^{(1)}\ln(\ell_1\ell_2) + c_n^{(2)}\ln(\ell_1 + \ell_2) + \cdots, \tag{156}$$

we find

$$-c_n^{(1)} = \sum_{k=0}^{n-1}\Delta_{\mathcal{T}_{k,n}^2} = \begin{cases} \frac{n^2-4}{12n} & n \text{ even} \\ \frac{n^2-1}{12n} & n \text{ odd} \end{cases} \tag{157}$$

and

$$-c_n^{(2)} = \sum_{k=0}^{n-1}(2\Delta_{\mathcal{T}_{k,n}} - \Delta_{\mathcal{T}_{k,n}^2}) = \begin{cases} \frac{n^2+2}{12n} & n \text{ even} \\ \frac{n^2-1}{12n} & n \text{ odd} \end{cases}, \tag{158}$$

which are the familiar results [70].

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
