# Peer review of "Twisted and untwisted negativity spectrum of free fermions"

_SciPost Physics, doi:SciPost Phys. 7, 037 (2019)_

## Round 1 · Referee Report · Anonymous · 2019-8-26

Strengths
- New and interesting results
- Readable and self-contained presentation
Report
The authors study the negativity spectrum of free fermions, defined
via the fermionic partial transpose (PT) of the density matrix.
This is constructed on a replicated spacetime manifold, by inserting
appropriate twist operators along the cuts (corresponding to the subsystems),
which encode the glueing conditions. It is pointed out, that there is a
freedom in the construction which can be used to define two different PT
operations, one of them being twisted. The authors then analyze the
so-called tail eigenvalue distribution for both of the PT density matrices.
This can be obtained by first calculating the corresponding moments,
and applying a Stieltjes transform, similarly as done for the simple
reduced density matrix.
The main results are as follows: while the twisted PT is a Hermitian
operator, having real (positive and negative) eigenvalues, the
untwisted PT is only pseudo-Hermitian and thus yields complex
eigenvalues. The latter ones are found to be distributed along
quantized phases of $\pi/3$ on the complex plane. The tail distribution
is then calculated along each of the branches for both twisted and
untwisted operators and the results are compared to lattice calculations
with a good agreement.
The results are interesting and the manuscript is nicely written
therefore I recommend its publication. I have only one question
which the authors may consider commenting on.
Requested changes
- The results for the tail distribution of the PT are given only for
adjacent intervals, although the moments for the disjoint case are
given explicitly in appendix B. Is the transformation to $P(\lambda)$
much more difficult to handle in this case?
- Stieltjes is misspelled before Eq. (21).
Author: Paola Ruggiero on 2019-09-19 [id 607]
(in reply to Report 1 on 2019-08-26)
We appreciate the referee’s positive feedback. We also thank the referee for pointing out the typo.
Regarding the referee’s question about the negativity spectrum of disjoint intervals: The method used in this paper can be applied to the case of two disjoint intervals similarly. However, one of our motivations to study adjacent intervals is that, in this case, the negativity calculations can be written in terms of three-point correlators of twist fields which are universal in CFTs. In contrast, the negativity of disjoint intervals involves four-point correlators of twist fields which in general depend on the full operator content of CFTs. Therefore, we think that our findings for the negativity spectrum of two adjacent intervals may apply to other fermionic CFTs although our calculations were done for free fermions.
We added a comment concerning this point in the conclusions of the paper.
Author: Paola Ruggiero on 2019-09-19 [id 606]
(in reply to Report 2 on 2019-08-30)We thank the referee for his feedback.
We just want to remark that the result of Ref.[110] mentioned by the referee
for the untwisted negativity spectrum of two adjacent intervals was actually overlooked there (and indeed the result given was misinterpreted). We comment about this point in a paragraph before eq.(70) on page 16, where we also provide its solution. More details were discussed in Appendix B.

---

## Round 1 · Referee Report · Anonymous · 2019-8-30

Report
The authors continue an exploration of modified measures of negativity for free, one dimensional fermions. In this manuscript, their emphasis is on the eigenvalue spectrum of the ``partial transpose'' of the density matrix. Partial transpose is placed in scare quotes because the naive notion of partial transpose has some undesirable features for free fermions -- for example it does not produce a Gaussian matrix. The authors here study two variants of the partial transpose, both of which involve also time reversal, and one of which is further partially twisted by $(-1)^{F}$ where $F$ is the fermion number operator. From a replica point of view, the usual Renyi negativity can be expressed as a sum over spin structures on a $n$-fold cover of the original manifold. The $n^{\rm th}$ moments of the two ``partial transposed'' density matrices considered here give two special terms in this sum, as emphasized in ref. [110].
Moving forward, it will be interesting to see if this modified notion of negativity remains useful in exploring disordered and interacting systems, in higher dimensions, and also in time dependent scenarios that involve quenches and thermalization.
I recommend publication.

---

## Editorial Decision

published